# Effortless, Simulation-Efficient Bayesian Inference using Tabular Foundation Models

**Julius Vetter[1,2] \* Manuel Gloeckler[1,2] \* Daniel Gedon[1,2] † Jakob H. Macke[1,2,3] †**

[1]Machine Learning in Science, University of Tübingen, Tübingen, Germany
[2]Tübingen AI Center, Tübingen, Germany
[3]Department Empirical Inference, Max Planck Institute for Intelligent Systems, Tübingen, Germany

## Abstract

Simulation-based inference (SBI) offers a flexible and general approach to performing Bayesian inference: In SBI, a neural network is trained on synthetic data simulated from a model and used to rapidly infer posterior distributions for observed data. A key goal for SBI is to achieve accurate inference with as few simulations as possible, especially for expensive simulators. In this work, we address this challenge by repurposing recent probabilistic foundation models for tabular data: We show how tabular foundation models—specifically TabPFN—can be used as pre-trained autoregressive conditional density estimators for SBI. We propose Neural Posterior Estimation with Prior-data Fitted Networks (NPE-PFN) and show that it is competitive with current SBI approaches in terms of accuracy for both benchmark tasks and two complex scientific inverse problems. Crucially, it often substantially outperforms them in terms of simulation efficiency, sometimes requiring orders of magnitude fewer simulations. NPE-PFN eliminates the need for selecting and training an inference network and tuning its hyperparameters. We also show that it exhibits superior robustness to model misspecification and can be scaled to simulation budgets that exceed the context size limit of TabPFN. NPE-PFN provides a new direction for SBI, where training-free, general-purpose inference models offer efficient, easy-to-use, and flexible solutions for a wide range of stochastic inverse problems.

## 1  Introduction

Simulation has long been a cornerstone of scientific inquiry [1, 2] and is becoming increasingly relevant as researchers tackle ever more complex scientific questions and systems [3]. Simulators often depend on parameters that are challenging or impossible to measure experimentally. Bayesian inference provides a general framework for identifying such parameters by estimating posterior distributions over parameters. However, classical methods such as Markov chain Monte Carlo (MCMC) require evaluations of the associated model likelihoods, which can be computationally demanding or prohibitive for complex numerical simulators.

The field of simulation-based inference (SBI), or likelihood-free inference, aims to address this challenge by enabling Bayesian inference without requiring access to likelihoods. The basic idea

---

\*Equal contribution.
†Joint supervision.
{firstname.lastname}@uni-tuebingen.de
Code available at https://github.com/mackelab/npe-pfn.

39th Conference on Neural Information Processing Systems (NeurIPS 2025).

of SBI is to train a neural network on synthetic data simulated from the model such that it can approximate the posterior distributions. In Neural Posterior Estimation (NPE, [4, 5]), a conditional density neural network—often, a normalizing flow [6, 7] or a diffusion model [8–10]—learns the conditional distribution of parameters given simulated data $p(\boldsymbol{\theta} \mid \boldsymbol{x})$. When evaluated on empirical observations $\boldsymbol{x}_o$, the network directly returns the posterior distribution $p(\boldsymbol{\theta} \mid \boldsymbol{x}_o)$. Similarly, other SBI methods use neural networks to represent likelihoods [11–13], likelihood ratios [14–17], or target several properties at once [18–21]. SBI has been used for scientific discovery in various domains, including astrophysics [22] and neuroscience [23].

Compared to classical Approximate Bayesian Computation (ABC, [24, 25]) or synthetic likelihoods methods [26], neural-network-based SBI methods scale better to complex, high-dimensional simulators [27]. In addition, SBI methods often *amortize* the computational burden: Once trained, the network provides fast inference across multiple observations.

Despite these advances, a key challenge for SBI methods remains: They typically require generating a large number of simulations as synthetic training data, rendering them impractical for expensive simulators. Recent work aims to increase simulation efficiency, e.g., by exploiting additional information or properties of the simulator [7, 28–31], utilizing low-fidelity simulations [32, 33], or using Bayesian neural networks [34–36]. So-called *sequential* approaches forego the amortization properties of the inference network and target simulations to particular observations [4–6, 37], but even then, simulation efficiency remains a challenge. In addition, users must select an appropriate SBI method and select and train an inference network with the associated hyperparameters—posing a barrier for inexperienced users.

These limitations raise a key question: Can we eliminate the high cost associated with training and tuning, and alleviate the need to have a large number of simulations by leveraging recent advances in foundation models? Here, we answer this question by repurposing a tabular foundation model—specifically Tabular Prior-data Fitted Networks v2 (TabPFN) [38]—as an inference engine for SBI. TabPFN is trained to perform in-context learning on tabular data (i.e., structured data organized in rows and columns), returning a prediction when being prompted with a training dataset and a test point. For both regression and classification, TabPFN was shown to perform exceptionally well, in particular in low data regimes. Crucially, TabPFN can also serve as a density estimator by applying it in an autoregressive manner, though prior work has only explored this in limited settings [38].

Here, we show that using TabPFN autoregressively allows for the estimation of complex, high-dimensional conditional densities, making it suitable for SBI. Based on this observation, we introduce Neural Posterior Estimation with Prior-data Fitted Networks (NPE-PFN), a variant of NPE that uses TabPFN as a pre-trained conditional density estimator, and therefore enables inference without any additional neural network training (Fig. 1). On benchmark tasks,

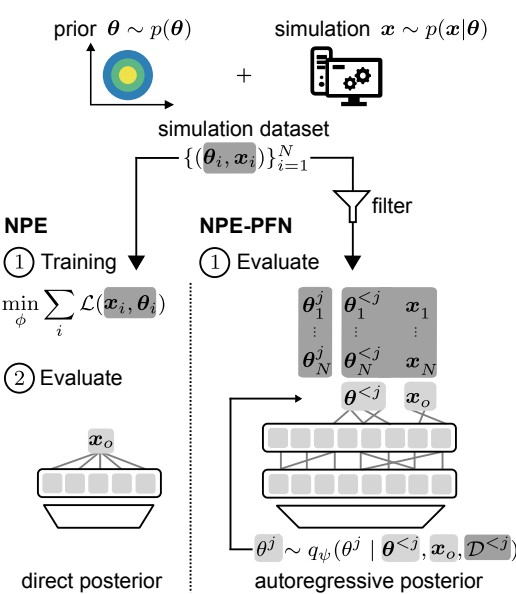

Figure 1: Comparison of NPE to NPE-PFN (ours): Both approaches use simulations sampled from the prior and simulator. In (standard) NPE, a neural density estimator is trained to obtain the posterior. In NPE-PFN, the posterior is evaluated by autoregressively passing the simulation dataset and observations to TabPFN.

NPE-PFN performs competitively with other SBI approaches and often outperforms them substantially, especially for small simulation budgets. Furthermore, we leverage data-filtering schemes that enable NPE-PFN to make use of large simulation budgets exceeding the context size of TabPFN and extend the approach to sequential inference settings. Finally, we show that NPE-PFN remains empirically robust under model misspecification and achieves strong results on two challenging scientific applications, requiring orders of magnitude fewer simulations than standard SBI approaches.

## 2 Neural Posterior Estimation with Prior-data Fitted Networks

### 2.1 Background

We consider a simulator with parameters $\boldsymbol{\theta} \in \mathbb{R}^{d_\theta}$ that stochastically generates samples $\boldsymbol{x} \in \mathbb{R}^{d_x}$, thus implicitly defining a likelihood $p(\boldsymbol{x} \mid \boldsymbol{\theta})$. Given a prior distribution $p(\boldsymbol{\theta})$ and observation $\boldsymbol{x}_o$, our aim is to infer the associated posterior distribution $p(\boldsymbol{\theta} \mid \boldsymbol{x}_o) \propto p(\boldsymbol{x}_o \mid \boldsymbol{\theta})p(\boldsymbol{\theta})$.

**Simulation-based inference.** We focus on Neural Posterior Estimation (NPE), a common approach to SBI: NPE approximates the target posterior distribution directly by training an inference network $q_\phi(\boldsymbol{\theta} \mid \boldsymbol{x})$ on a dataset of parameter–data pairs $\mathcal{D} = \{(\boldsymbol{\theta}_i, \boldsymbol{x}_i)\}_{i=1}^N$ sampled from a prior $p(\boldsymbol{\theta})$ and a simulator with implicit likelihood $p(\boldsymbol{x} \mid \boldsymbol{\theta})$. The inference network is trained to maximize the log-likelihood and learns to approximate the corresponding posterior distribution [4, 39]. Flexible conditional neural density estimators, such as normalizing flows [40–42], are typically used as inference networks. At inference time, NPE provides an amortized posterior approximation, allowing for rapid evaluations.

**Tabular foundation models.** Recent foundation models for tabular data have shown strong performance on regression and classification tasks via in-context learning, in which predictions for new inputs are made by conditioning the model on a context dataset [43]. We focus on Prior-data Fitted Networks (PFNs) [44, 45], specifically TabPFN [38], a state-of-the-art transformer-based PFN foundation model [46] for probabilistic classification and regression on tabular data. It has been pre-trained on synthetic datasets $\{(y_i, \boldsymbol{x}_i)\}_{i=0}^M$ with up to $M = 2048$ samples from randomly generated structural causal models by maximizing the likelihood $q_\psi(y_0 \mid \boldsymbol{x}_0, \mathcal{T})$, where $\mathcal{T} = \{(y_i, \boldsymbol{x}_i)\}_{i=1}^M$ is the in-context dataset. After training, TabPFN has been tested to perform well up to a feature dimension of $d_x \leq 500$ and a context size of $N \leq 10^4$ [38].

At inference, TabPFN conditions on a given in-context dataset $\mathcal{T}'$ and test point $\boldsymbol{x}_o$ to estimate the distribution $p(y \mid \boldsymbol{x}_o) \approx q_\psi(y \mid \boldsymbol{x}_o, \mathcal{T}')$ over the target variable $y$. Importantly, in its basic version, TabPFN is limited to *univariate* targets ($y \in \mathbb{R}$ for regression, $y \in \{1, \ldots, n\}$ for classification). Thus, TabPFN is an in-context density estimator for one-dimensional densities. We perform inference over a model parameter $\theta \in \mathbb{R}$ by setting $y = \theta$ to estimate the posterior distribution $p(\theta \mid \boldsymbol{x}_o) \approx q_\psi(\theta \mid \boldsymbol{x}_o, \mathcal{D})$ given an observation $\boldsymbol{x}_o$ and a dataset of simulations $\mathcal{D}$. However, to estimate full posterior distributions over multiple model parameters $\boldsymbol{\theta} \in \mathbb{R}^{d_\theta}$, conditional density estimation in high dimensions is required.

### 2.2 TabPFN for simulation-based inference

To perform simulation-based inference with TabPFN, we repurpose it as a general conditional density estimator. The key to estimating high-dimensional densities with TabPFN is to use it autoregressively, i.e., to use it to sequentially predict the next dimension of the data with the previously processed dimensions and covariates in context.

In SBI, given a prior $p(\boldsymbol{\theta})$ and simulator $p(\boldsymbol{x} \mid \boldsymbol{\theta})$, the multivariate posterior distribution $p(\boldsymbol{\theta} \mid \boldsymbol{x}_o)$ for observation $\boldsymbol{x}_o$ can be decomposed as

$$p(\boldsymbol{\theta} \mid \boldsymbol{x}_o) \approx \prod_{j=1}^{d_\theta} q_\psi(\theta^j \mid \boldsymbol{\theta}^{<j}, \boldsymbol{x}_o, \mathcal{D}^{<j})$$

where $\mathcal{D}^{<j} = \{\theta_i^j, [\boldsymbol{\theta}_i^{<j}, \boldsymbol{x}_i]\}$ with $<j$ denoting the vector from index 1 up to, but not including $j$. Thus, using an arbitrary but fixed order of the parameter dimensions, one evaluation of the posterior $p(\boldsymbol{\theta} \mid \boldsymbol{x}_o)$ requires $d_\theta$ evaluations of TabPFN, making the prediction more expensive as the dimensionality increases. However, unlike standard NPE methods, this approach bypasses model fitting entirely and directly evaluates the posterior. We refer to our proposed approach as NPE-PFN, in contrast to standard NPE, which requires training an inference network, e.g., a normalizing flow (Fig. 1, pseudocode for NPE-PFN in Appendix Alg. 1).

### 2.3 Increasing the effective context size

Standard NPE can approximate arbitrarily complex conditional densities, given enough simulations and a sufficiently expressive inference network [4]. In contrast, NPE-PFN is limited by TabPFN's

context size of approximately $10^4$ samples, beyond which additional data yields diminishing performance gains. While we expect NPE-PFN to be particularly powerful for *small* simulation counts, it is nevertheless desirable to also be able to work with larger simulation budgets using in-context learning.

**Filtering simulations based on relevance.** To overcome the limited context size, previous work proposed to use a subset of the training data containing only the nearest neighbors of a given test point as the context dataset [47]. Here, we show that this procedure, and generalizations of it, are sound from a Bayesian perspective when estimating the full posterior distribution. To this end, we exploit a key property of (neural) posterior estimation: The posterior can be estimated using only samples that are very close to the observation $\boldsymbol{x}_o$. More specifically, for any non-negative function $f_{\boldsymbol{x}_o}$ that satisfies $f_{\boldsymbol{x}_o}(\boldsymbol{x}_o) > 0$, we can reweigh the joint distribution $p(\boldsymbol{x}, \boldsymbol{\theta})f_{\boldsymbol{x}_o}(\boldsymbol{x})$ without affecting the posterior distribution at convergence (details in Appendix Sec. B.2). This property allows us to *filter* simulations based on relevance without biasing the estimation. Choosing the filter as $f_{\boldsymbol{x}_o}(\boldsymbol{x}) = \mathbb{I}(d(\boldsymbol{x}, \boldsymbol{x}_o) < \epsilon)$, where $d(\cdot, \cdot)$ is a suitable distance metric and $\epsilon > 0$ is a threshold, effectively recovers an ABC-like selection scheme, where only simulations close to $\boldsymbol{x}_o$ are retained. In practice, $\epsilon$ is set adaptively to include the $N_{\text{filter}}$ closest simulations, ensuring that the full context is used. Here, we consider the Euclidean distance $d(\boldsymbol{x}_o, \boldsymbol{x}) = \|\boldsymbol{x} - \boldsymbol{x}_o\|_2$ after standardizing each feature dimension. As noted above, this filtering based on nearest neighbors has previously been explored and is also referred to as *localization* or *retrieval* [47–49]. This filtering principle also underlies various SBI methods, such as the aforementioned ABC [24] or calibration kernels in NPE training [5].

Filtering for simulations that are relevant or close to a given observation makes the (context) dataset dependent on that observation. For classical SBI methods, this step would require re-training the density estimator for each new observation, thus breaking amortization. However, NPE-PFN remains amortized: Since it is training-free, inference for new observations requires only a computationally negligible filtering step over a fixed set of simulations. Finally, filtering only works for *conditional* density estimation problems such as (simulation-based) Bayesian inference, which is our focus here. However, we also show how (Appendix Sec. B.4) this approach can be extended to *unconditional* density estimation on a larger number of samples. In the unconditional case, we partition the data space into regions that can be solved with local contexts.

**Embedding data.** Similarly to context size, the feature size can be a limitation, particularly for high-dimensional observations. This limitation can be handled either by performing inference using human-crafted summary statistics (as commonly done in ABC) or using (pre-trained) embedding networks [5, 50]. In principle, NPE-PFN allows for training an embedding network end-to-end (e.g., [51] in the context of classification), but we here investigate pre-training an embedding network independently of the density estimator (Appendix Sec. B.3).

## 2.4 Truncated sequential NPE-PFN

To reduce the number of simulations, *sequential* SBI methods have been developed that acquire simulations adaptively in rounds for one observation $\boldsymbol{x}_o$. We focus on Truncated Sequential NPE (TSNPE) [37], which mitigates bias by exploiting the (approximate) invariance of the posterior under modifications outside of its high-density regions (HDR). Formally, let $\text{HDR}_p^{1-\alpha} := \{\boldsymbol{\theta} \mid p(\boldsymbol{\theta} \mid \boldsymbol{x}_o) > k_\alpha\}$, where $k_\alpha$ denotes the density threshold containing $\alpha\%$ of the probability mass. Then, given a truncated prior $\tilde{p}(\boldsymbol{\theta}) = \mathbb{I}(\boldsymbol{\theta} \in \text{HDR}_p^{1-\alpha})p(\boldsymbol{\theta})$, the posterior will be approximately preserved for $\boldsymbol{x}_o$. In TSNPE, a density estimator is trained in rounds. Starting with the full prior in the first round, the trained posterior approximation $q(\boldsymbol{\theta} \mid \boldsymbol{x}_o)$ is used to estimate $\text{HDR}_p^{1-\alpha} \approx \text{HDR}_q^{1-\alpha}$ and to truncate the prior to the HDR of the posterior approximation for the next round. In each subsequent round, the newly truncated prior is used to generate new simulations adaptively and refine $q(\boldsymbol{\theta} \mid \boldsymbol{x}_o)$ through continued training. Notably, training becomes redundant by replacing the density estimator with the TabPFN regressor, to which we will refer as TSNPE-PFN. In this training-free setting, the main computational burden shifts to truncating the prior distribution. Prior truncation can be achieved either by approximate approaches, such as sampling-importance resampling, or exactly by rejection sampling from the prior, i.e., checking whether a prior sample falls within the HDR. However, the latter approach requires repeated density estimation with the TabPFN regressor to evaluate $q(\boldsymbol{\theta} \mid \boldsymbol{x}_o)$, which can be costly to do autoregressively, a problem we address next.

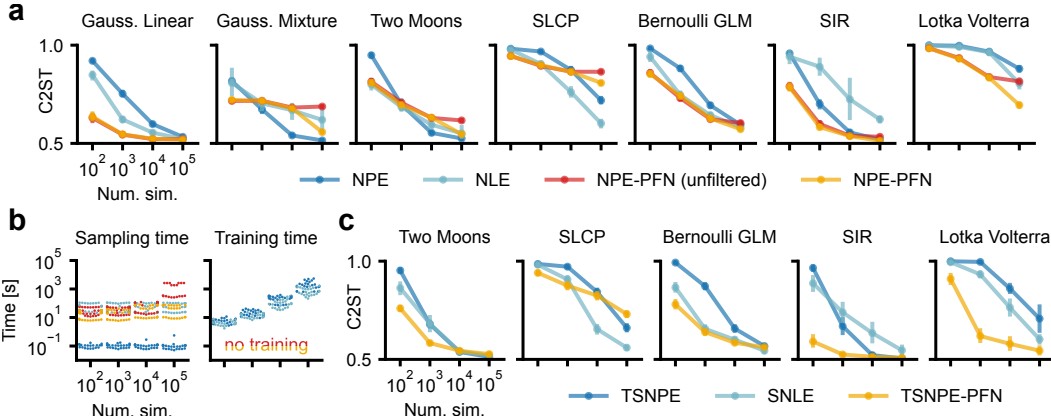

Figure 2: **SBI benchmark results for amortized and sequential NPE-PFN. (a)** C2ST for NPE, NLE, and NPE-PFN across ten reference posteriors (lower is better); dots indicate averages and bars show 95% confidence intervals over five independent runs. **(b)** Average time to generate $10^4$ posterior samples and (if applicable) training time. **(c)** C2ST for sequential methods.

For rejection sampling, we need to evaluate $q(\boldsymbol{\theta} \mid \boldsymbol{x}_o)$ for a large number of prior samples $\boldsymbol{\theta} \sim p(\boldsymbol{\theta})$. In practice, the posterior is often much narrower than the prior and a large percentage of these samples will be rejected. This high rejection rate necessitates a fast, approximate approach for density evaluation, which we will construct using the "ratio-trick" [52]. We first sample an initial set of posterior samples $\{\boldsymbol{\theta}_i\}_i$ from the NPE-PFN posterior once and assign these samples the class label $y = 1$. We then contrast these samples with samples from a uniform distribution $\mathcal{U}(\boldsymbol{\theta}; \boldsymbol{\theta}_{min}, \boldsymbol{\theta}_{max})$ to which we assign the class label $y = 0$. The Bayes optimal classifier for this classification problem is given by

$$P_{\boldsymbol{x}_o}(y = 1 \mid \boldsymbol{\theta}) = \frac{p(\boldsymbol{\theta} \mid \boldsymbol{x}_o)}{p(\boldsymbol{\theta} \mid \boldsymbol{x}_o) + \mathcal{U}(\boldsymbol{\theta}; \boldsymbol{\theta}_{min}, \boldsymbol{\theta}_{max})}, \quad \text{thus,} \quad p(\boldsymbol{\theta} \mid \boldsymbol{x}_o) = \frac{P_{\boldsymbol{x}_o}(y = 1 \mid \boldsymbol{\theta})}{1 - P_{\boldsymbol{x}_o}(y = 1 \mid \boldsymbol{\theta})},$$

within the bounds $\boldsymbol{\theta}_{min}, \boldsymbol{\theta}_{max}$. After autoregressively sampling from the desired posterior to construct the training dataset, this approach, which we will refer to as *ratio-based* density evaluation, requires only a single forward pass using the TabPFN classifier and is, therefore, significantly faster than autoregressive density evaluation, especially as the parameter dimension $d_\theta$ or the number of required density evaluations grows (pseudocode for TSNPE-PFN in Appendix Alg. 2).

## 3 Experiments

To assess NPE-PFN, we conduct experiments on synthetic SBI benchmark tasks and real data, covering scenarios from low to high-dimensional data and including cases with model misspecification.

### 3.1 Amortized and sequential benchmark performance

We evaluate NPE-PFN on various tasks from the SBI benchmark [27], which provides ground truth posterior samples for 10 observations for each task. We measure posterior sample quality using the classifier two-sample test (C2ST, 53). A C2ST accuracy of 0.5 indicates that the approximate posterior exactly matches the ground truth posterior, as the classifier fails to distinguish between the two sample sets. In contrast, an accuracy of 1.0 reflects a strong mismatch between the estimated and true posterior. As baselines, we compare against flow-based NPE and NLE provided by the SBI library [54, 55]. For sequential baselines, we use truncated sequential NPE (TSNPE [37]) and sequential NLE (SNLE [11]). Posterior estimates are computed across four simulation budgets, ranging from $10^2$ to $10^5$ simulations. Beyond $10^4$ simulations, NPE-PFN incorporates filtering, whereas an alternative unfiltered variant, denoted *NPE-PFN (unfiltered)*, extends the context size beyond the TabPFN-recommended limit. For budgets $\leq 10^4$, both variants are identical.

**Amortized inference.** For $10^2$ simulations, NPE-PFN substantially outperforms NLE and NPE on all but one task (Fig. 2a,b). For $10^3$ or $10^4$ simulations, NPE-PFN is generally competitive, with

the exception of a few tasks where some baselines outperform it. For example, NLE outperforms NPE-PFN on the simple-likelihood-complex-posterior (SLCP) task, as NPE-PFN, like NPE, directly targets the posterior instead of the simpler likelihood. However, NPE-PFN performs substantially better than both baselines on tasks such as SIR or Lotka-Volterra for all simulation budgets. With $10^5$ simulations, the difference between NPE-PFN and its unfiltered variant becomes evident. The extended context does not improve performance for larger simulation budgets, while the filtered NPE-PFN variant achieves significantly better C2ST accuracies, matching or surpassing the baselines. While NPE-PFN is training-free, inference speed depends on the number of simulations, and the dimensionality of the parameter and observation spaces. For the benchmark tasks, we observe an inference speed that is comparable to NLE (Fig. 2b), but orders of magnitude slower than NPE, which is near-instantaneous. We perform a careful analysis of the inference speed of NPE-PFN across all relevant variables for users to make an informed decision in their respective application (Appendix Sec. D.3, Tab. D-2).

The advantage of NPE-PFN over baseline methods remains for other tasks outside the SBI benchmark suite (Appendix Sec. D.1, Fig. D-1), as well as compared to other baseline methods such as NRE, NPE ensembles (Appendix Sec. D.2, Fig. D-2), and even when performing extensive hyperparameter optimization for NPE (Appendix Fig. D-4). Furthermore, NPE-PFN posterior estimates are well-calibrated (Fig. D-3).

In cases where data dimension $d_x$ outgrows the supported feature dimension and learned summary statistics are required, the end-to-end trained NPE, however, outperforms NPE-PFN with independently pre-trained summary statistics (Appendix Sec. D.4, Fig. D-5). In addition, we ablate the effect of the filter size indicating that the default size is (close to) optimal for all considered tasks (Appendix Sec. D.6, Fig. D-7). Finally, we vary the autoregressive order, in which the parameter dimensions are sampled, and find no noticeable effect compared to the default order (Appendix Sec. D.8).

**Sequential inference.** We adapt the setup in Deistler et al. [37] to evaluate TSNPE-PFN. Rejection sampling is performed using the fast, approximate ratio-based densities (Sec. 2.4, Appendix Sec. D.5, Fig. D-6 for a comparison between autoregressive and ratio-based density evaluation). We equally divide the simulation budget into 10 rounds, with proposals truncated to the $1 - \varepsilon$ highest density region, where $\varepsilon = 10^{-3}$. TSNPE-PFN excels with small simulation budgets (Fig. 2c). On the SIR task, it reaches close to optimal C2ST accuracy with only $10^2$ simulations. At larger budgets, TSNPE-PFN maintains a strong performance, particularly on the Lokta-Volterra task, which is highly challenging for other SBI methods. TSNPE-PFN benefits from NPE-PFN's ability to generate high-quality posterior estimates with few simulations, allowing it to acquire more relevant simulations in early rounds.

**Summary.** NPE-PFN provides accurate posterior estimates, especially for small simulation budgets. With filtering, NPE-PFN is also competitive for larger simulation budgets. As an in-context learning method, it requires no training on specific simulations and offers inference speeds comparable to other baselines, though runtime scales with the number of simulations and the dimensionality of the parameters and observations. This flexibility makes NPE-PFN applicable to a wide range of setups and simulators. We observe NPE-PFN performs particularly well in tasks with underlying graphical structure and conditional (in-)dependencies, which are similar to the structural causal models used in TabPFN's pre-training. In addition to posterior estimation with NPE-PFN, we evaluate the unconditional high-dimensional density estimation capabilities of TabPFN on UCI datasets [56], showing that it is more data-efficient than a neural spline flow (Appendix Sec. D.9, Fig. D-10).

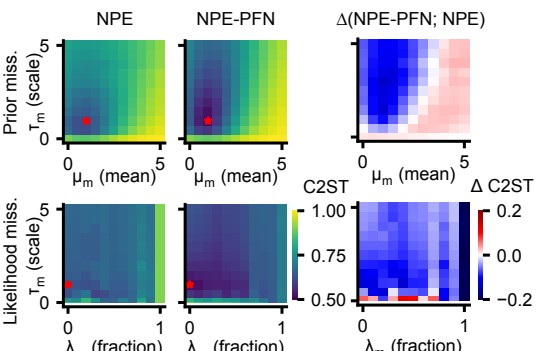

Figure 3: **Robustness under misspecification.** C2ST against a well-specified ground truth posterior. The red star marks the well-specified model. **Rows:** Prior and likelihood misspecification. **Columns:** NPE, NPE-PFN, and their difference in terms of C2ST accuracy (darker blue indicates that NPE-PFN is better).

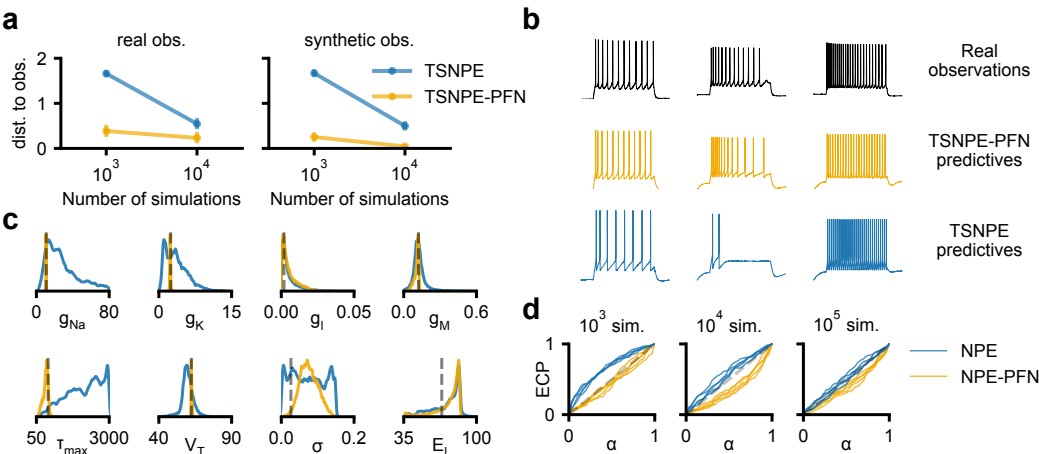

Figure 4: **Posterior inference for observations from the Allen cell type database.** **(a)** Average distance to observation in standardized space of summary statistics for both the real and synthetic observations. **(b)** Posterior predictive simulations for real observations using TSNPE-PFN and TSNPE. **(c)** Posterior marginals for one synthetic observation; TSNPE-PFN marginals are substantially more constrained for several parameters. **(d)** Simulation-based calibration for NPE-PFN and NPE on the HH simulator.

## 3.2 Impact of misspecification

To investigate how NPE-PFN handles misspecification, we use the 2D Gaussian means misspecification benchmark from Schmitt et al. [57], considering both prior and likelihood misspecification. The ground truth is given by the prior $\mu \sim \mathcal{N}(\mathbf{0}, \boldsymbol{I})$ and likelihood $x_i \sim \mathcal{N}(\mu, \boldsymbol{I})$. Prior misspecification uses the prior $\mu \sim \mathcal{N}(\mu_m, \tau_m \boldsymbol{I})$ while keeping the true likelihood; Likelihood misspecification uses the true prior with the likelihood $x_i \sim \lambda_m \text{Beta}(2, 5) + (1 - \lambda_m)\mathcal{N}(\mu, \tau_m \boldsymbol{I})$, where $\tau_m \in \mathbb{R}^+$, $\lambda_m \in [0, 1]$. We evaluate the C2ST of predicted posteriors against the reference posterior distribution of the well-specified data using $10^2$ simulations (details in Appendix Sec. 3.2).

Both NPE-PFN and NPE degrade as the level of misspecification increases, with prior misspecification leading to more severe performance degradation than likelihood misspecification (Fig. 3). However, NPE-PFN achieves better C2ST values than NPE across a broader range of misspecification levels. In addition, NPE-PFN remains insensitive to various feature and noise distributions (Appendix Sec. D.7, Fig. D-8). Finally, we find that the performance of NPE-PFN on the SBI benchmark tasks is invariant to the choice of the autoregressive order (Appendix Sec. D.8, Fig. D-9).

Thus, the trend observed in the SBI benchmark extends to these simple but misspecified tasks, on which NPE-PFN also consistently outperforms NPE. Next, we evaluated whether this performance gap also holds for scientific tasks with real data, on which the simulator is potentially misspecified.

## 3.3 Single-Compartment Hodgkin-Huxley Model

We evaluate the performance of (TS)NPE-PFN on a challenging inference task, considering the Hodgkin-Huxley (HH) simulator of single-neural voltage dynamics [58, 59]. Following Gonçalves et al. [60], the simulator has eight parameters and seven summary statistics based on the voltage trace. We infer posteriors for 10 real observations from the Allen cell type database [61], for which inference is known to be challenging [62–64]. We evaluate inference quality via posterior predictives and the average Euclidean distance between predictives and real observations in a standardized summary statistics space. In addition, for each real observation, we create an associated synthetic, and therefore well-specified observation, where the ground truth parameter is known, allowing for direct comparison between inferred posteriors and true values.

TSNPE-PFN yields posterior predictives that closely match *real observations* with a simulation budget of $10^4$ (Fig. 4a,b). It outperforms TSNPE, which struggles to produce realistic predictives, especially at smaller budgets. For *synthetic observations*, TSNPE-PFN again achieves better predictive accuracy

with fewer simulations, with the average distance approaching zero for a simulation budget of $10^4$. In parameter space, TSNPE-PFN posteriors are tightly concentrated around the true parameters, with significant improvement over TSNPE (Fig. 4c). In the *amortized setting*, NPE-PFN shows good calibration for $10^3$ and $10^5$ simulations but is slightly overconfident for $10^4$ (Fig. 4d). However, (TS)NPE-PFN achieves significantly better prediction quality and yields tight posteriors around the true parameters, making it more sensitive to miscalibration.

These results demonstrate TSNPE-PFN's superior simulation efficiency, achieving the same predictive quality as TSNPE with an order of magnitude fewer simulations. This efficiency makes TSNPE-PFN particularly well-suited for inference with real, potentially misspecified simulators under limited simulation budgets.

### 3.4 Pyloric Network Model

Finally, we evaluate TSNPE-PFN on a high-dimensional simulator of the pyloric network in the stomatogastric ganglion (STG) of the crab *Cancer Borealis*, a well-studied circuit generating rhythmic activity [60, 65, 66]. This model consists of three neurons, each with eight membrane conductances, interconnected by seven synapses, resulting in a 31-dimensional parameter space. The simulated voltage traces are summarized using 15 established statistics. A key challenge in this setting is the extreme sparsity of valid simulations. Specifically, 99% of parameter samples from the prior yield implausible voltage traces, preventing the computation of summary statistics [67].

For this reason, earlier work relied on millions of simulations for an amortized NPE-based posterior approximation (18 million in Gonçalves et al. [60]). The simulation count could later be reduced by restricting the prior to *valid* simulation using an additional classifier [67]. In addition, several sequential algorithms have been developed to address this problem [18, 37, 68], the latest of which achieves good posterior predictives using $1.5 \cdot 10^5$ simulations.

To perform sequential inference on the pyloric network with TSNPE-PFN, we use a restricted prior based on the TabPFN classifier. Due to the high dimensionality of the parameter space, rejection sampling is prohibitive. Therefore, we use sampling importance resampling with an oversampling factor of 10. In the first round, we use $5 \cdot 10^3$ simulations and run TSNPE-PFN for 45 rounds, adding $10^3$ simulations per round for a total of $5 \cdot 10^4$ simulations (details in Appendix Sec. C.4).

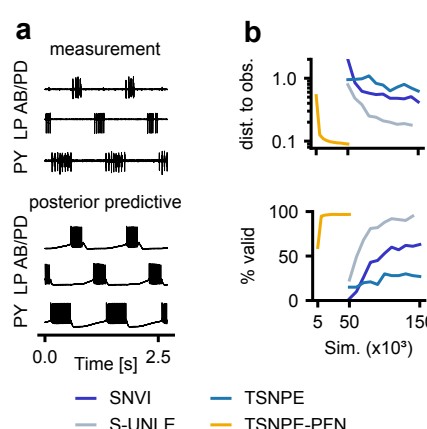

The generated voltage traces from posterior mean estimates of TSNPE-PFN closely resemble true measurements (Fig. 5). With more than $50\%$ valid simulations at just $5 \cdot 10^3$ simulations and approximately $99\%$ validity for fewer than $5 \cdot 10^4$ simulations, TSNPE-PFN outperforms all comparison methods, which either required substantially larger simulation budgets to reach comparable performance or did not reach it at all. Furthermore, the autoregressive evaluation order again has no noticeable effect on the resulting performance (Appendix Sec. C.4).

Figure 5: **Results on the pyloric simulator.** **(a)** Voltage traces from the experimental measurement (top) and a posterior predictive simulated using the posterior mean from TSNPE-PFN as the parameter (bottom). **(b)** Average distance (energy scoring rule) to observation and percentage of *valid* simulation from posterior samples; compared to experimental results obtained in Glaser et al. [68].

Our results demonstrate that TSNPE-PFN can infer posteriors with substantially improved simulation efficiency, making it a powerful approach for complex high-dimensional models with a limited simulation budget.

## 4 Discussion

We have shown here that pre-trained foundation models—specifically, TabPFN—can be effectively repurposed to perform accurate training-free simulation-based inference. Together with autoregressive

sampling for high-dimensional posterior distributions, filtering to increase context size, and fast density evaluation using the density ratio, NPE-PFN provides a flexible inference method that can be effortlessly applied to many different inference tasks. Across several benchmarks and two real-world tasks, NPE-PFN consistently estimates accurate posteriors with particularly strong results for small simulation budgets, often requiring orders of magnitude fewer simulations than existing SBI methods. These results establish foundation models and in-context learning as a promising direction for training-free inference in scientific applications.

**Related work.** Our work relates to **meta-learning** [69], where models generalize across tasks by acquiring shared inductive biases. Meta learning has been applied to amortized (Bayesian) inference [70, 71]. More specifically, however, NPE-PFN is an instance of **in-context learning** (ICL), where task adaptation is achieved solely by conditioning pre-trained models on different inputs [43]. ICL was popularized by LLMs [72], but its scope has expanded beyond language, with work showing that transformers can learn various function classes in context [73, 74] and work introducing prior fitted networks [38, 44, 45]. NPE-PFN extends this direction by enabling ICL for simulation-based Bayesian inference. We note that this does not require training the inference networks on *any* SBI tasks, but rather works well using an "off-the-shelf" model trained for classification and regression.

In addition to NPE-PFN, multiple methods have been proposed to perform **ICL for Bayesian inference** [75–77]. These approaches parameterize posteriors using, for example, Gaussians, flows, or mixtures and require fixed input and parameter dimensionality and, to date, are only trained on a specific, limited class of probabilistic models. In contrast, NPE-PFN leverages autoregressive modeling over parameter dimensions, enabling zero-shot generalization without retraining across inference problems with varying shapes.

Finally, several works have proposed **extensions for TabPFN** beyond its original setting, targeting new data modalities, larger context sizes, or higher-dimensional features. To address TabPFN's limited context size, retrieval-based approaches select informative subsets of the training data [47–49, 78], including nearest-neighbor-based filtering [47–49], which we adapt in NPE-PFN for Bayesian inference. Crucially, we demonstrate that this filtering does not bias the posterior, as the filter step can be interpreted as a calibration kernel [5]. Other strategies include data distillation [79, 80], boosting weak learners [81], feature-wise ensembling [82], encoders and bagging [51], as well as non-autoregressive extensions to time series forecasting [83] and tabular data generation [84]. TabICL [85] proposes architectural changes to scale to datasets with over $10^5$ points. NPE-PFN departs from these approaches by repurposing TabPFN for posterior inference through filtering and fast evaluation of densities.

**Limitations.** While NPE-PFN eliminates the need for simulation-specific training, it inherits properties of TabPFN. TabPFN has a soft limit in maximum feature size ($\leq 500$) and context length ($\leq 10^4$). We present methods to overcome these limitations, i.e, filtering and embedding nets. However, as the number of simulations and the need for density evaluations increase, the performance advantage of NPE-PFN over standard methods such as NPE diminishes. Furthermore, while embedding networks allow NPE-PFN to be used for high-dimensional observations, they require separate pre-training. In addition, modeling high-dimensional parameter spaces beyond TabPFN's maximum feature size of 500 dimensions remains challenging. Finally, like other in-context methods, NPE-PFN is more expensive than flow-based NPE, a challenge exacerbated by the autoregressive estimation of posteriors. However, unlike other in-context methods for Bayesian inference, NPE-PFN does not require training and hyperparameter tuning, making it suitable for exploratory workflows or non-expert users. In addition, by requiring substantially fewer simulations, NPE-PFN accelerates the overall SBI workflow. Thus, while we expect NPE-PFN to be particularly useful for exploratory workflows or expensive simulators, inference methods based on trained density estimators might still be advantageous in settings where large numbers of simulations are available, or where bespoke embedding networks and fast sampling are required, such as in some applications in astrophysics [22]. An interesting direction for future work is to transfer NPE-PFN's advantages to existing, faster inference techniques (e.g., via student–teacher distillation).

**Conclusion.** We presented NPE-PFN, a method that repurposes the tabular foundation model TabPFN for training-free simulation-based Bayesian inference. NPE-PFN is competitive with—and often outperforms—existing methods, especially for small simulation budgets. NPE-PFN is a simple-to-use and simulation-efficient tool for SBI, with the potential to become the go-to method for novice users, exploratory workflows, and applications with constrained simulation budgets.

## Acknowledgments and Disclosure of Funding

We thank Michael Deistler and Katharina Eggensperger for feedback on the manuscript, specifically Michael Deistler for suggesting to filter simulations and Jonas Beck for the implementation of the pyloric simulator. We thank all members of the Mackelab for discussions and feedback on the manuscript. This work was funded by the German Research Foundation (DFG) under Germany's Excellence Strategy – EXC number 2064/1 – 390727645 and SFB 1233 'Robust Vision' (276693517), the European Union (ERC, DeepCoMechTome, 101089288), the "Certification and Foundations of Safe Machine Learning Systems in Healthcare" project funded by the Carl Zeiss Foundation and the Boehringer Ingelheim AI & Data Science Fellowship Program. JV and MG are members of the International Max Planck Research School for Intelligent Systems (IMPRS-IS).

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

# Appendix Table of Contents

# A    Software and computational resources

Our implementation of NPE-PFN is based on PyTorch [86] and the TabPFN [38] library. We use the SBI library [54] to run the SBI baselines, and hydra [87] for experiment and hyperparameter management. Code to use NPE-PFN and reproduce the results is available at `https://github.com/mackelab/npe-pfn`.

We use a mix of Nvidia 2080TI, A100, and H100 GPUs to obtain the results related to NPE-PFN. Some results, such as those for the unfiltered NPE-PFN variant together with a simulation budget of $10^5$ require the use of GPUs with high VRAM due to the large context used. SBI baselines were run on 8 CPU cores, as normalizing flow training and sampling only benefit from GPUs for large sample sizes and dimensionalities.

# B    Additional method details

## B.1    Pseudocode: NPE-PFN and TSNPE-PFN

In Alg. 1 and Alg. 2, we provide pseudocode for NPE-PFN and TSNPE-PFN, respectively. Note that NPE-PFN is training-free. In practice, this means that training consists of storing the training data $\mathcal{D} = \{(\boldsymbol{\theta}_i, \boldsymbol{x}_i)\}_{i=1}^{N}$ for in-context processing during inference.

---

**Algorithm 1** NPE-PFN

---

**Require:** Observation $\boldsymbol{x}_o$, simulations $\mathcal{D}_{\text{full}} = \{(\boldsymbol{\theta}_i, \boldsymbol{x}_i)\}_{i=1}^{N}$, TabPFN regressor $q_\psi^{\text{reg}}(\cdot \mid \cdot)$, filter size $N_{\text{filter}}$

1: **if** $N_{\text{filter}} < N$ **then**                                              ▷ if filtering is active
2:       $d_i = d(\boldsymbol{x}_o, \boldsymbol{x}_i)$                                        ▷ compute filter distances
3:       $\mathcal{D} \leftarrow \{(\boldsymbol{\theta}_i, \boldsymbol{x}_i) : i \in \text{argtop}_{N_{\text{filter}}}(d_i)\}$       ▷ select $N_{\text{filter}}$ closest simulations
4: **else**
5:       $\mathcal{D} \leftarrow \mathcal{D}_{\text{full}}$
6: **end if**
7: **for** $j = 1, \ldots, d_{\boldsymbol{\theta}}$ **do**                                      ▷ for all parameter dimension $d_{\boldsymbol{\theta}}$
8:       $\mathcal{D}^{<j} = \{\theta_i^j, [\boldsymbol{\theta}_i^{<j}, \boldsymbol{x}_i]\}$                        ▷ build context data set
9:       $\theta^j \sim q_\psi^{\text{reg}}(\theta^j \mid \boldsymbol{\theta}^{<j}, \boldsymbol{x}_o, \mathcal{D}^{<j})$       ▷ sample one parameter dimension from TabPFN
10: **end for**
11: **return** $\boldsymbol{\theta} = [\theta_1, \theta_2, \ldots, \theta_{d_{\boldsymbol{\theta}}}]^{\top}$                             ▷ posterior sample

---

## B.2    Filtering to increase effective context size

To allow NPE-PFN to make use of more than $10^4$ simulations, we filter the simulations based on their relevance to a given observation. In the following, we show that this procedure does not bias the estimation of the posterior and discuss details and possible extensions.

For a given prior $p(\boldsymbol{\theta})$ and likelihood $p(\boldsymbol{x} \mid \boldsymbol{\theta})$, the joint distribution is given by $p(\boldsymbol{x}, \boldsymbol{\theta}) = p(\boldsymbol{x} \mid \boldsymbol{\theta})p(\boldsymbol{\theta})$ and the posterior distribution is given by

$$p(\boldsymbol{\theta} \mid \boldsymbol{x}) = \frac{p(\boldsymbol{x} \mid \boldsymbol{\theta})p(\boldsymbol{\theta})}{p(\boldsymbol{x})} = \frac{p(\boldsymbol{x}, \boldsymbol{\theta})}{\int p(\boldsymbol{x}, \boldsymbol{\theta}') \, d\boldsymbol{\theta}'}. \tag{B-1}$$

Formally, a filter is a non-negative function $f_{\boldsymbol{x}_o}(\boldsymbol{x})$ based on some observation $\boldsymbol{x}_o$, where we require $f_{\boldsymbol{x}_o}(\boldsymbol{x}_o) > 0$. Crucially, reweighing the joint distribution by the filter does not affect the posterior. More specifically, with the reweighted joint $\tilde{p}(\boldsymbol{x}, \boldsymbol{\theta}) \propto p(\boldsymbol{x}, \boldsymbol{\theta})f_{\boldsymbol{x}_o}(\boldsymbol{x})$, we have

$$\tilde{p}(\boldsymbol{\theta} \mid \boldsymbol{x}) = \frac{\tilde{p}(\boldsymbol{x}, \boldsymbol{\theta})}{\int \tilde{p}(\boldsymbol{x}, \boldsymbol{\theta}') \, d\boldsymbol{\theta}'} = \frac{f_{\boldsymbol{x}_o}(\boldsymbol{x})p(\boldsymbol{x}, \boldsymbol{\theta})}{\int f_{\boldsymbol{x}_o}(\boldsymbol{x})p(\boldsymbol{x}, \boldsymbol{\theta}') \, d\boldsymbol{\theta}'} = p(\boldsymbol{\theta} \mid \boldsymbol{x}), \tag{B-2}$$

as $f_{\boldsymbol{x}_o}(\boldsymbol{x})$ cancels out due to its independence from $\boldsymbol{\theta}$.

In theory, any function $f$ satisfying these properties can be used to filter for relevant simulations. In this work, we only consider an ABC-like filter, where $f_{\boldsymbol{x}_o}(\boldsymbol{x}) = \mathbb{I}(d(\boldsymbol{x}, \boldsymbol{x}_o) < \epsilon)$ is the indicator

---

**Algorithm 2** TSNPE-PFN with rejection sampling

---

**Require:** Observation $\boldsymbol{x}_o$, simulator $p(\boldsymbol{x} \mid \boldsymbol{\theta})$, prior $p(\boldsymbol{\theta})$, TabPFN classifier $q_\psi^{\mathrm{cls}}(\cdot \mid \cdot)$, TabPFN regressor $q_\psi^{\mathrm{reg}}(\cdot \mid \cdot)$, number of rounds $R$, number of posterior samples for ratio $M$, simulations per round $N_r$, truncation fraction $\alpha$

    — *Initialize dataset* —
1: Draw $N_r$ prior samples $\boldsymbol{\theta}_i \sim p(\boldsymbol{\theta})$
2: Simulate samples $\boldsymbol{x}_i \sim p(\boldsymbol{x} \mid \boldsymbol{\theta}_i)$
3: Initialize $\mathcal{D} = \{(\boldsymbol{\theta}_i, \boldsymbol{x}_i)\}_{i=1}^{N_r}$

4: **for** $r = 1, \ldots, R$ **do**
        — *Construct fast ratio-based density estimator* —
5:     Draw $M$ posterior samples $\boldsymbol{\theta}_j^{\mathrm{post}} \sim q(\boldsymbol{\theta} \mid \boldsymbol{x}_o)$ using Alg. 1 with $\mathcal{D}$ as context
6:     Assign class $y = 1$ to $\boldsymbol{\theta}_j^{\mathrm{post}}$ and $y = 0$ to $\boldsymbol{\theta}_j^{\mathrm{uni}} \sim \mathcal{U}(\boldsymbol{\theta}_{\min}, \boldsymbol{\theta}_{\max})$
7:     Define classifier dataset $\mathcal{D}_{\mathrm{cls}} = \{(\boldsymbol{\theta}_j, y_j)\}_{j=1}^{2M}$ combining $\boldsymbol{\theta}_j^{\mathrm{post}}$ and $\boldsymbol{\theta}_j^{\mathrm{uni}}$
8:     With $\mathcal{D}_{\mathrm{cls}}$ as context, $q_\psi^{\mathrm{cls}}$ can be used to estimate posterior density:

$$q_{\mathrm{cls}}(\boldsymbol{\theta} \mid \boldsymbol{x}_o) = \frac{q_\psi^{\mathrm{cls}}(y = 1 \mid \boldsymbol{\theta}, \mathcal{D}_{\mathrm{cls}})}{1 - q_\psi^{\mathrm{cls}}(y = 1 \mid \boldsymbol{\theta}, \mathcal{D}_{\mathrm{cls}})}.$$

        — *Estimate high-density region (HDR)* —
9:     qs $= \{q_{\mathrm{cls}}(\boldsymbol{\theta}_j^{\mathrm{post}} | \mathbf{x}_o)$ for $j = 1, \ldots, M\}$         ▷ evaluate posterior densities
10:    $k_\alpha = \mathrm{quantile}(\mathrm{qs}, \alpha)$         ▷ estimate HDR threshold

        — *Rejection sample from truncated prior* —
11:    $D_r = \emptyset$
12:    **while** $|D_r| < N_r$ **do**
13:        $\boldsymbol{\theta} \sim p(\boldsymbol{\theta})$         ▷ sample from prior
14:        **if** $q_{\mathrm{cls}}(\boldsymbol{\theta} \mid \mathbf{x}_o) \geq k_\alpha$ **then**         ▷ accept if in HDR
15:           $\mathbf{x} \sim p(\mathbf{x} \mid \boldsymbol{\theta})$
16:           $D_r \leftarrow D_r \cup \{(\boldsymbol{\theta}, \mathbf{x})\}$
17:        **end if**
18:    **end while**
19:    $\mathcal{D} = \mathcal{D} \cup \mathcal{D}_r$
20: **end for**

    — *Sample final samples* —
21: Draw posterior samples $\boldsymbol{\theta}_i \sim q(\boldsymbol{\theta} \mid \boldsymbol{x}_o)$ using Alg. 1 with $\mathcal{D}$ as context
22: **return** $\{\boldsymbol{\theta}_i\}_i$

---

function that evaluates to 1, when some distance $d(\boldsymbol{x}, \boldsymbol{x}_o)$ between the observation $\boldsymbol{x}_o$ simulation $\boldsymbol{x}$ is smaller than some threshold $\epsilon > 0$, and is 0 otherwise. As described in Sec. 2.3, we use the Euclidean distance in feature-wise standardized observation space and choose $\epsilon$ adaptively to include the $10^4$ closest simulations.

The autoregressive manner in which we use TabPFN for density estimation allows for more complex filter designs that consider not only the observation $\boldsymbol{x}_o$, but also already processed parameter dimensions. In particular, it is possible to select parameter-simulation pairs $(\boldsymbol{\theta}, \boldsymbol{x})$ based on their relevance to the parameter dimensions $<j$ *and* observation $\boldsymbol{x}_o$, when using TabPFN to estimate $p(\theta^j \mid \boldsymbol{x}_o, \boldsymbol{\theta}^{<j}) = q_\psi(\theta^j \mid \boldsymbol{x}_o, \boldsymbol{\theta}^{<j}, \mathcal{D}^{<j})$. We do not investigate this approach here and leave the design of more complex (autoregressive) filters to future work.

## B.3   Embedding networks for high-dimensional data

For NPE, it is typical to employ an embedding net $e = f_\phi(\boldsymbol{x})$ depending on the case of high-dimensional data. The embedding network can be end-to-end trained together with the associated

conditional density estimator $q_\phi(\boldsymbol{\theta} \mid f_\phi(\boldsymbol{x}))$ and helps to identify lower-dimensional "sufficient" statistics, which the density estimator can more effectively handle [5]. The embedding net is usually chosen to follow existing properties of the data at hand; i.e., for image-like simulation, one might employ a convolutional neural network.

In principle, this end-to-end approach is directly applicable to NPE-PFN, but it would require differentiation through the TabPFN model, which can be expensive. We instead aim to obtain lower dimensional embedding, i.e., "summary features" independent from the pre-trained model. This idea is similar to feature selection schemes usually employed for TabPFN [51, 88].

While various approaches exist for addressing this problem, we investigate the method proposed by Chen et al. [50], which focuses on maximizing the mutual information between the parameters $\boldsymbol{\theta}$ and their corresponding embeddings $e = f_\phi(\boldsymbol{x})$, with data $\boldsymbol{x} \sim p(\boldsymbol{x} \mid \boldsymbol{\theta})$. The goal is to derive embeddings that serve as "sufficient" statistics for accurately inferring parameters from observed data. However, directly maximizing mutual information is generally intractable, leading to the development of proxy measures.

We employ the distance-correlation proxy, which aligns the distances between embeddings with the distances between their associated parameters. Specifically, we optimize the following loss function:

$$\mathcal{L}(\phi) = \frac{\mathbb{E}_{p(\boldsymbol{x},\theta)p(\boldsymbol{x}',\theta')}\left[d(\theta, \theta')\, d(f_\phi(\boldsymbol{x}), f_\phi(\boldsymbol{x}'))\right]}{\sqrt{\mathbb{E}_{p(\theta)p(\theta')}[d(\theta, \theta')^2]\,\mathbb{E}_{p(\boldsymbol{x})p(\boldsymbol{x}')}[d(f_\phi(\boldsymbol{x}), f_\phi(\boldsymbol{x}'))^2]}} \tag{B-3}$$

We refer to this approach as NPE-PFN-Infomax and present results on using NPE-PFN together with pre-trained embedding nets in Sec. D.4.

### B.4 Unconditional density estimation

As outlined in Sec. 2.1, TabPFN is designed as a *conditional* density estimator for one-dimensional densities. For higher-dimensional conditional density estimation, TabPFN can be applied autoregressively. To adapt TabPFN for unconditional density estimation, random Gaussian noise is added as the first dimension. The subsequent conditional density estimation is then performed in the same way as for conditional tasks [38].

For conditional density estimation (e.g., posterior estimation), filtering provides an effective method to optimize the context given the corresponding condition. However, this strategy does not generalize to unconditional density estimation. One solution is to manually introduce a conditional structure. Specifically, we partition the support of the target density into independent clusters and estimate each component separately. Formally, we express the target density as a mixture model

$$p(\boldsymbol{x}) = \sum_{i=1}^{n} p(\boldsymbol{x} \mid c_i)\, p(c_i), \tag{B-4}$$

where $p(c_i)$ is the probability of sampling cluster $c_i$, and $p(\boldsymbol{x} \mid c_i)$ is the cluster-specific density estimated via NPE-PFN. This mixture model approach effectively increases the usable context size with the number of clusters. In this work, we apply $k$-means clustering, which has been explored for TabPFN by Xu et al. [78] outside of density estimation. Other methods to introduce conditional structure, such as those proposed by Li et al. [89], represent promising directions for future research. We present results on unconditional density estimation in Sec. D.9.

## C Experimental details

### C.1 SBI benchmark simulators for amortized and sequential inference

To benchmark NPE-PFN and the sequential version TSNPE-PFN, we use a set of tasks from the SBI benchmark [27]. This benchmark contains several challenging inference tasks with nonlinear dependencies and multi-modal posteriors. We provide an overview of the dimensionality of parameters and observations for all benchmark tasks (Tab. C-1). As described in Sec. 3.1, we measure the quality of the inferred posteriors with a classifier-two-sample-test (C2ST). For the classifier, we use a random forest with the default hyperparameters provided by the SBI library [54].

Table C-1: **Overview on the SBI benchmark tasks.** For details on the specification of the prior and the exact simulator equations, we refer to Lueckmann et al. [27, Appendix T].

| Task Name | $\dim \boldsymbol{\theta}$ | $\dim \boldsymbol{x}$ | Notes |
|---|---|---|---|
| Gaussian Linear | 10 | 10 | Mean inference in 10-D Gaussian with fixed covariance |
| Gaussian Mixture | 2 | 2 | Common mean of two 2-D Gaussians with varying covariance |
| Two Moons | 2 | 2 | Bimodal, two moon-shaped posterior structure |
| SLCP | 5 | 8 | Simple Gaussian likelihood, complex posterior |
| Bernoulli GLM | 10 | 10 | GLM with Bernoulli observations, smoothness prior |
| SIR | 2 | 10 | Epidemic SIR model, inference of contact and recovery rates |
| Lotka-Volterra | 4 | 20 | Predator-prey dynamics, species interaction parameters |

The baseline methods NPE, NLE, NRE and their sequential variants are implemented using the SBI library. All methods use the default normalizing flows (neural spline flow [42] for NPE, masked autoregressive flow [41] for NLE) with hyperparameters suggested by the SBI library. Training was performed using the Adam optimizer [90] with a batch size of 200 and a learning rate of $5 \cdot 10^{-4}$. Training was stopped early based on the validation loss, as evaluated on a held-out set containing 10% of the available simulations. For the NPE ensembles, five different estimators are trained with different random seeds to obtain a mixture of equally weighted estimators.

In all experiments, we use the default version of the TabPFN classifier or regressor for (TS)NPE-PFN, with no changes to hyperparameters such as the softmax temperature. All runtimes for NPE-PFN (Fig. 2b) were obtained using an Nvidia A100 GPU, where possible. For the unfiltered variant of NPE-PFN, an H100 GPU was used for the large context containing $10^5$ simulations.

## C.2 Misspecification simulator

Here, we detail the experimental setup of Sec. 3.2. The misspecification benchmark from Schmitt et al. [57] is given by

| | | | | |
|---|---|---|---|---|
| **Truth** | Prior: | $\boldsymbol{\mu} \sim \mathcal{N}(\mathbf{0}, \boldsymbol{I})$, | Likeli.: | $\boldsymbol{x}_i \sim \mathcal{N}(\boldsymbol{\mu}, \boldsymbol{I})$ | (C-5) |
| **Prior** | Prior: | $\boldsymbol{\mu} \sim \mathcal{N}(\boldsymbol{\mu}_m, \tau_m \boldsymbol{I})$, | Likeli.: | $\boldsymbol{x}_i \sim \mathcal{N}(\boldsymbol{\mu}, \boldsymbol{I})$ | (C-6) |
| **Likeli.** | Prior: | $\boldsymbol{\mu} \sim \mathcal{N}(\mathbf{0}, \boldsymbol{I})$, | Likeli.: | $\boldsymbol{x}_i \sim \lambda \text{Beta}(2, 5) + (1 - \lambda)\mathcal{N}(\boldsymbol{\mu}, \tau_m \boldsymbol{I})$ | (C-7) |

where $\mu_m \in \mathbb{R}$ as mean shift, $\tau_m \in \mathbb{R}^+$ as standard deviation scaling, and $\lambda \in [0, 1]$ as the fraction of how many samples are from the true normal or a different beta distribution.

For the experiments, we ran 100 simulations for three seeds over all combinations of $(\mu_m, \tau_m, \lambda)$. NPE and NPE-PFN are trained to predict posterior samples. These samples are compared to the reference posterior distribution using C2ST. This reference posterior distribution is defined with respect to the well-specified distribution. Thus, this task is designed to evaluate the robustness of the model under prior or likelihood misspecification.

## C.3 Single-compartment Hodgkin-Huxley simulator

To perform inference on the observations from the Allen cell types database [61], we use a single-compartment Hodkin-Huxley model following the one proposed in Pospischil et al. [59]. This model has four types of conductances (sodium, delayed-rectifier potassium, slow voltage-dependent potassium, and leak) and a total of eight parameters. Following Gonçalves et al. [60], who previously used this model for simulation-based inference, we compute seven summary statistics on the simulated and observed action potentials. These are spike count, mean, and standard deviation of the resting potential, and the first four voltage moments (mean, standard deviation, skew, and kurtosis). We also use the same uniform prior as in Gonçalves et al. [60]. For sequential inference with TSNPE and TSNPE-PFN, we use a total of $10^3$ or $10^4$ simulations, equally divided over five rounds. For both methods, we sample from the truncated proposal using rejection sampling, taking into account the $1 - \varepsilon$ highest density region with $\varepsilon = 10^{-3}$.

Here, we provide posterior predictive samples for all 10 observations from the Allen cell types database using TSNPE-PFN (Fig. D-11) and TSNPE (Fig. D-12), complementing the results presented in the main text (Fig. 4). Importantly, both inference and the computation of the average predictive

distance are performed on the summary statistics and not directly on the simulations or observations. Thus, features beyond those encoded by the summary statistics are not captured. For TSNPE-PFN, the posterior predictives closely match with the observations in most cases, with a few exceptions such as obs. 3 and obs. 6. In contrast, flow-based TSNPE yields posterior predictives that are less similar to the observations, with a few exceptions such as obs. 1 and obs. 8. These qualitative results further support the quantitative results reported in the main text, demonstrating that TNSPE-PF performs more reliable inference with fewer simulations.

## C.4  Pyloric Network Model

A key challenge in this setting is the extreme sparsity of valid simulations. In particular, 99% of parameter samples from the prior yield implausible voltage traces, preventing the computation of summary statistics. For this reason, previous work has relied on millions of simulations for an amortized NPE-based posterior approximation (18 million in Gonçalves et al. [60]). The simulation count could later be reduced to 9 million by restricting the prior to *valid* simulations with an additional classifier [67]. Moreover, several sequential algorithms have been developed to tackle this problem. First, SNVI was proposed, which learns both a likelihood and a variational approximation of the posterior. Notably, the likelihood model was only based on "valid" simulations and corrected by a classifier that predicts whether parameters are valid. This approach made it possible to produce good posterior samples with $3.5 \cdot 10^5$ simulations [18]. This number was reduced even further to only $1.5 \cdot 10^5$ simulations by S-UNLE, which made use of energy-based models and Langevin dynamic MCMC [68]. Later, this approach was extended to SNPE approximations, specifically using TSNPE [37].

Similarly, we want to use only *valid* simulations in the context of NPE-PFN and therefore adaptively truncate the prior using a TabPFN classifier. This approach is based on the intuition that the posterior should place little or no probability mass on regions in parameter space that are likely to lead to invalid simulations [18, 37, 67]. Specifically, as simulations accumulate over rounds, we randomly fill the context of the TabPFN classifier with up to $5 \cdot 10^3$ valid and invalid simulations. The prior is then truncated to $\tilde{p}_{\text{val}}(\boldsymbol{\theta}) \propto \mathbb{I}(P(\text{valid} \mid \boldsymbol{\theta}) > c)p(\boldsymbol{\theta})$, where we choose $c = 0.3$. We then use TSNPE-PFN together with an approximate importance resampling scheme instead of rejection sampling for efficiency. Specifically, we oversample by a factor of $K = 10$ and obtain $10^4$ samples, which are then resampled to $10^3$ samples based on their importance weight $w(\boldsymbol{\theta}) = \tilde{p}_{\text{val}}(\boldsymbol{\theta})/q_\phi(\boldsymbol{\theta} \mid \boldsymbol{x}_o)$. For evaluation, we exactly follow Glaser et al. [68] by computing the percentage of valid simulations as well as the energy score in each round. Note, however, that the overall experimental setting is not perfectly identical as we use a faster implementation of the pyloric network simulator based on Jaxley [91].

Previous sequential methods started with an initial simulation budget of $5 \cdot 10^4$ prior simulations to acquire a set of $\sim 500$ valid simulations. Due to the effectiveness of NPE-PFN on small simulation budgets, we start with only $5 \cdot 10^3$ simulations, thus acquiring only $\sim 50$ valid simulations in the first round. From there, we acquire $10^3$ new simulations in each round, updating both NPE-PFN and the restricted classifier with the additional data. We stop after 45 rounds for a total of $5 \cdot 10^4$ simulations. Despite this limited budget, TSNPE-PFN quickly converges to a posterior that produces a high number of valid simulations that closely match the observed measurement. After the final round, we obtain a valid percentage of 96.83% and an energy score of 0.0899 using only $5 \cdot 10^4$ simulations—the starting point of the other methods—and still surpass their final performance (Fig. 5). The evaluation presented in the main text only examines the fidelity of the posterior predictives. Therefore, we include a visualization of the full posterior distribution after the final round (Fig. D-13), which shows broad marginal distributions and similar features as the posteriors reported in previous work [18, 37, 60, 68].

In terms of runtime, most of the computation time is spent on sampling from the restricted prior and running simulations. During the first few rounds, the process is faster (about 2 minutes) due to the smaller context size in TSNPE-PFN. Once the context limit is reached, the time to propose the next $10^3$ parameters stabilizes at about 7 minutes, using an Nvidia H100 GPU. Notably, the performance of previous state-of-the-art methods on this task is surpassed after only 10 rounds—an hour of runtime (including simulation time)—which is substantially faster than any previously proposed methods [18, 68].

We choose the default order as the autoregressive order for sampling. The total number of possible permutations in this task is greater than $10^{33}$, thus finding the optimal permutation in this space is highly challenging. To gain some insight, we perform an ablation study on 50 distinct random permutations using the context dataset identified by the final-round posterior. We compute the presented metrics with $10^3$ simulations. We found that the energy score varies between 0.081 - 0.094 and the valid rates between 96% - 98% (5% and 95% quantile each). These ranges match closely with the results reported in Fig. 5 and demonstrate that there is no significant effect for the autoregressive order in this experiment. Appendix Sec. D.8 reports the same conclusion across all benchmark tasks.

## D    Additional experiments

### D.1    Additional tasks

Here we extend the SBI benchmark experiments (Sec. 3.1) with additional inference tasks from other scientific fields or interesting synthetic tasks used in previous work.

Specifically, the additional tasks include simulators from physics (Weinberg, Stellar Streams), computer science (M/G/1 queue) [16], as well as "structured" synthetic tasks (Tree, HMM) [21]. Out of these, the stellar streams simulator stands out as particularly computationally demanding ($> 30$ min. per simulation per CPU core; with two-dimensional parameters, and 199 dimensional data). For this reason, we investigate the performance only up to $10^4$ simulations.

The tasks in Hermans et al. [16] do not come with a ground truth posterior. We therefore evaluate them only via the average log posterior density (Fig. D-1a). Overall, the results on these additional tasks again demonstrate that, for a small simulation budget, NPE-PFN significantly outperforms NPE. Specifically, NPE-PFN strongly outperforms NPE on tasks with sparse conditional dependencies, i.e., Tree, HMM, Streams, in which it is one to three orders of magnitudes more simulation-efficient than NPE. The stellar streams task is a good example use-case, where this efficiency is particularly beneficial: Attaining 100 simulations is feasible on a consumer-grade CPU (10 cores) in 5 hours, and NPE-PFN achieves a comparable performance to NPE with $10^4$ simulations, which would require 20 days. In addition, we investigated the calibration of NPE-PFN for which we observed a similar trend (Fig. D-1b on a subset of tasks). Finally, we performed a direct comparison to reference posteriors where available. We also extended the HMM task to 50 parameter and data dimensions as an additional test for high-dimensional yet "structured" parameter spaces (Fig. D-1c). Here, NPE-PFN generally outperformed NPE.

The HMM example specifically illustrates how NPE (or, more generally, conditional density estimation) fundamentally struggles with dimensionality, independent of the complexity of the simulation process. One approach to address this challenge is to incorporate suitable inductive biases or constraints (such as known conditional dependencies) into the estimation network architecture [21, 30, 92]. The inductive bias from TabPFN as used in NPE-PFN seems to be especially beneficial in such cases. TabPFN was trained exclusively on synthetic datasets generated from random structural causal models (SCMs), which by construction exhibit (sparse) conditional (in-)dependencies. We therefore hypothesize that pretraining on SCMs enables TabPFN to automatically detect and exploit such (in-)dependencies directly from data. A detailed investigation of these mechanisms would be an interesting direction for future work.

### D.2    SBI benchmark with more baseline methods

Here, we extend the SBI benchmark experiments (Sec. 3.1) by including two additional baselines: neural ratio estimation (NRE) from Durkan et al. [14] and an ensemble of NPE models, denoted NPE (Ensemble) following Hermans et al. [16]. Ensemble models are known to improve predictive performance [93] and, in the context of SBI, to improve simulation-based calibration (SBC) [16]. We evaluate all methods under the same experimental conditions as before (Sec. 3.1), using five equally weighted ensemble members for NPE (Ensemble).

We present results in terms of C2ST and SBC metrics (Fig. D-2, extending the results from the main text (Fig. 2). For SBC, we use the Error of Diagonal (EoD) to quantify the deviation from perfect calibration. NRE performs similarly or worse on all tasks, while NPE (Ensemble) matches standard NPE in terms of C2ST and achieves the best calibration of all baselines. NPE-PFN is comparable to NPE (Ensemble) in terms of calibration on most tasks, while often outperforming it in

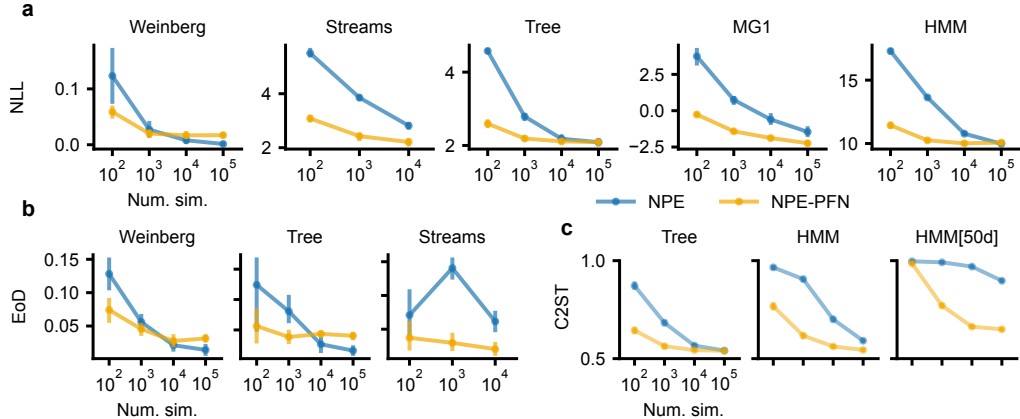

Figure D-1: **Results on additional tasks.** (**a**) Performance under the average negative log likelihood metric (NLL) across five new tasks for NPE and NPE-PFN. (**b**) Calibration error on a subset of the new tasks. (**c**) Performance in C2ST against reference posteriors on tasks for which reference are available. We also investigate a different parameterization of the HMM tasks by increasing dimensionality. Note that the simulation budget of the Streams tasks only goes up to $10^4$ simulations.

predictive accuracy and simulation efficiency. Calibration plots (Fig. D-3) corroborate the EoD-based calibration summary with fine-grained information.

These results underscore that NPE-PFN not only achieves strong predictive performance but also provides well-calibrated posteriors that match or exceed ensembles. While we do not directly compare NPE-PFN with the work of Delaunoy et al. [36], who present a SBI method for small simulation budgets using Bayesian neural networks, their method reports modest gains over ensembles, whereas NPE-PFN consistently outperforms them—further emphasizing the effectiveness of our approach with a limited simulation budget.

Throughout this manuscript, the comparisons of NPE-PFN with baseline methods are performed using the sbi library's default hyperparameters [55]. Here, we additionally perform hyperparameter optimization of our main baseline method NPE. For each task and simulation budget, we conduct a random search for flow (flow type, number of flow layers, dimensionality of hidden flow layers) and optimization (batch size, learning rate) hyperparameters. In each setting, we limit the computing time to ten hours. For seven tasks and four simulation budgets, this results in a total computing time of 280 hours.

Compared to NPE with the default settings, NPE (Sweep) achieves better performance in some tasks, particularly with smaller simulation budgets (Fig. D-4). This is due to the shorter training times with $10^2$ and $10^3$ simulations, enabling the random search to cover large parts of the search space. For $10^4$ and $10^5$ simulations, no or only minor improvements can be observed. Further improvements for larger budgets would require more computing time and better hyperparameter optimization algorithms, such as Bayesian optimization. Despite the substantial computing time per setting, the optimized NPE (Sweep) still lags behind NPE-PFN in terms of performance for most tasks with smaller budgets (Fig. D-4). Overall, these results again demonstrate NPE-PFN's strong default performance, providing an easy-to-use method that is competitive with, or even superior to, baseline methods where intensive hyperparameter optimization was performed. This strong default performance is particularly beneficial in the sequential setting, where hyperparameter optimization poses significant challenges due to long inference times, changing training conditions across rounds, and the difficulty of selecting suitable metrics for optimization.

### D.3 Inference speed

While NPE-PFN is completely training-free, the autoregressive use of TabPFN requires computational effort during inference (Fig. 2b). The inference speed of NPE-PFN depends on three parameters: 1) the number of simulations passed as the in-context dataset; 2) the dimensionality of the observation space (i.e., the number of features); and 3) the dimensionality of the parameter space (i.e., the number of parameters).

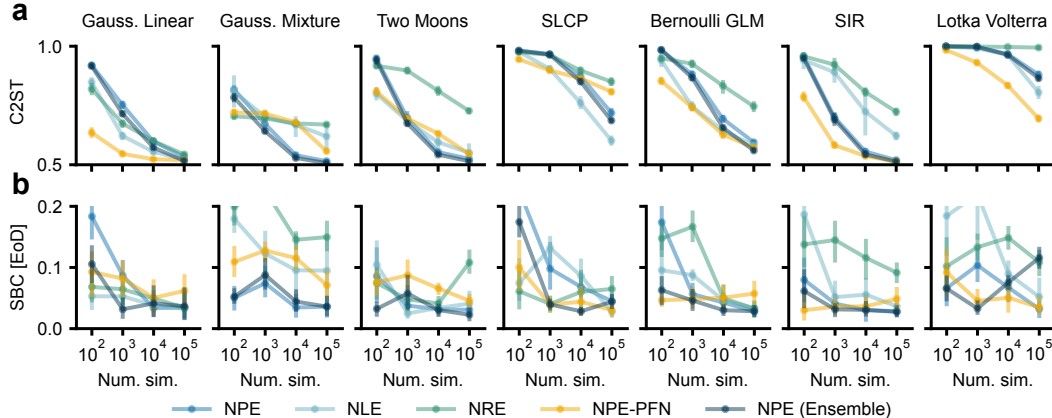

Figure D-2: **Extended SBI benchmark results for amortized NPE-PFN. (a)** Extension of Fig. 2a with more baseline methods, namely NRE and NPE (Ensemble). **(b)** Mean absolute error between the calibration curves and the diagonal (EoD), with 0 indicating perfect calibration. Full calibration curves in Fig. D-3.

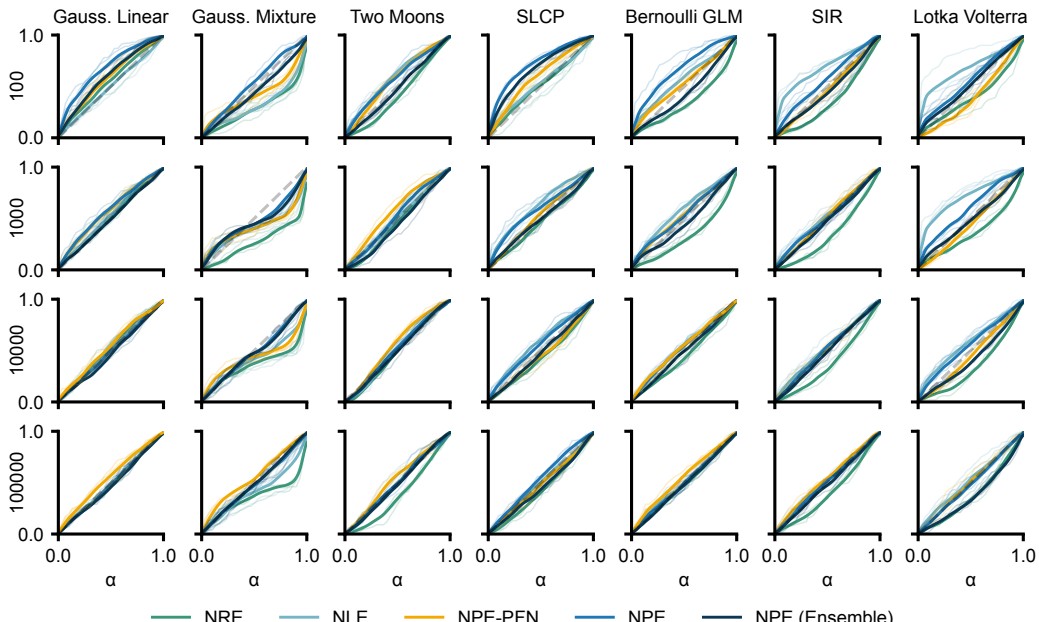

Figure D-3: **Full calibration curves for all tasks.** For each task (columns) and each simulation budget (rows), we plot the associated mean calibration curve (bold) as well as 3 individual runs (transparent, thin).

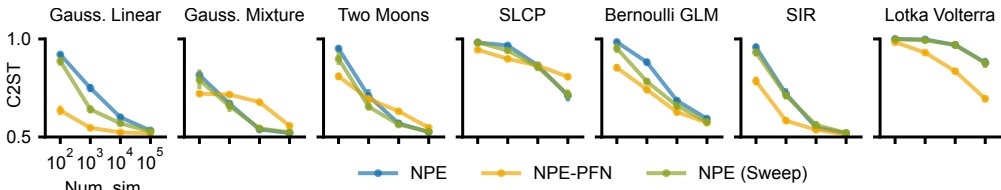

Figure D-4: **SBI benchmark results with NPE hyperparameter optimization.** C2ST for NPE-PFN, NPE with default settings, and NPE (Sweep), where hyperparameter optimization was performed. Specifically flow and optimization hyperparameters are optimized separately for each benchmark task and simulation budget using ten hours of random search.

Here, we provide a thorough examination of the inference speed of NPE-PFN across different simulation budgets and varying dimensionalities of the parameter and observation spaces (Tab. D-2). All results were obtained using an Nvidia A100 GPU. In general, as the value of any of the three parameters increases, inference becomes slower. The dimensionality of the parameter space has the strongest impact on overall inference speed, since the autoregressive use of TabPFN necessitates the reprocessing of the adapted context for each additional parameter dimension. For the largest parameter and observation dimensions and a context of $10^4$ simulations, inference can take several minutes. However, this can be alleviated by filtering; for example, by selecting the $10^3$ most relevant simulations (see Sec. D.6 for details).

Table D-2: **Inference speed of NPE-PFN.** Inference speed to sample a batch of $10^4$ samples from the posterior measured in *seconds* (on an Nvidia A100 GPU). Rows show increasing dimensionality of the parameter space, columns show increasing dimensionality of the observation space, stratified by different simulation budgets.

| $\theta$ \ $x$ | $10^2$ Sim. | | | | | $10^3$ Sim. | | | | | $10^4$ Sim. | | | | |
|---|---|---|---|---|---|---|---|---|---|---|---|---|---|---|---|
| | 2 | 4 | 8 | 16 | 32 | 2 | 4 | 8 | 16 | 32 | 2 | 4 | 8 | 16 | 32 |
| 2 | 6 | 4 | 5 | 6 | 7 | 7 | 6 | 5 | 9 | 8 | 7 | 8 | 10 | 14 | 23 |
| 4 | 11 | 10 | 10 | 11 | 13 | 10 | 11 | 11 | 13 | 17 | 15 | 19 | 24 | 32 | 48 |
| 8 | 19 | 23 | 22 | 27 | 30 | 23 | 23 | 24 | 27 | 35 | 35 | 40 | 49 | 66 | 101 |
| 16 | 45 | 47 | 47 | 53 | 63 | 49 | 52 | 54 | 61 | 77 | 89 | 99 | 116 | 152 | 221 |
| 32 | 97 | 103 | 107 | 122 | 137 | 115 | 116 | 125 | 141 | 171 | 251 | 272 | 307 | 370 | 516 |

## D.4   Embedding networks for high-dimensional data

We evaluate the performance of **NPE-PFN-Infomax** on high-dimensional simulation-based inference tasks, specifically focusing on a spatial SIR model [16] and an extended Lotka-Volterra task [94]. In the case of the spatial SIR task, where the true posterior is intractable, we rely on indirect metrics such as negative log-likelihood (NLL), TARP [95], and SBC for evaluation. NPE-PFN-Infomax leverages compressed embeddings learned via an infomax objective to distill key information from high-dimensional observations, which are then used as inputs to NPE-PFN.

For the spatial SIR task, we employ a 2D CNN with four convolutional layers consisting of 16, 32, 64, and 128 channels, each with a kernel size of 5 and max pooling of size 2. The resulting feature maps are flattened and passed through a two-layer MLP to produce a 10-dimensional summary statistic. For the extended Lotka-Volterra task (300-dimensional time series), we adopt a similar architecture using a 1D CNN with a kernel size of 3 and an output embedding of 16 dimensions.

Our results (Fig. D-5) indicate that such embedding networks—whether pre-trained or jointly trained—can be used to extend the applicability of NPE-PFN to high-dimensional observations. However, in most cases, this approach does not outperform an end-to-end trained NPE baseline. Interestingly, TabPFN without an embedding network (when applicable) often achieves better performance, suggesting that the compression step, independent of NPE-PFN, may be suboptimal. One possible explanation is that TabPFN, having been trained on causally structured data, can exploit underlying dependencies to improve inference accuracy. In contrast, compressing such highly structured data into low-dimensional summaries may discard critical information, limiting the effectiveness of inference.

## D.5   Comparison of autoregressive and ratio-based density evaluation

In Sec. 2.4, we present a method for fast density evaluation based on density ratios, requiring only a single forward pass through the TabPFN classifier. While this approach offers a substantial computational advantage, it is inherently approximate, raising the question of how it compares to the autoregressive density evaluation. To answer this question, we compare the two methods on two tasks—the 2D Two Moons tasks and the 10D Gaussian Linear task—across varying simulation budgets. In each case, we evaluate the density of posterior samples from the trained model given an observation, using 5 random seeds. To construct the dataset for the ratio estimator, we draw $5 \cdot 10^3$ samples each from the posterior of interest and the uniform base distribution, which is the default setting.

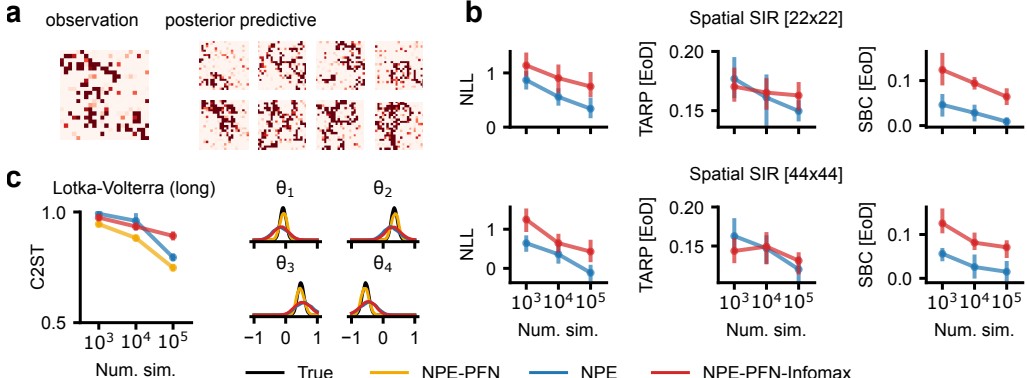

Figure D-5: **High-dimensional data with embedding nets. (a)** Visualization of observations in the spatial SIR task alongside posterior predictive samples obtained using NPE-PFN-Infomax. **(b)** Performance evaluation across high-dimensional variants of the spatial SIR task over 5 independent runs, reported in terms of the negative log-likelihood (NLL) of the true parameter and the area off the diagonal in TARP and SBC calibration analyses. **(c)** An instance of the Lotka-Volterra task with an extended time series (300 dimensions), evaluated using NPE-PFN and NPE-PFN-Infomax.

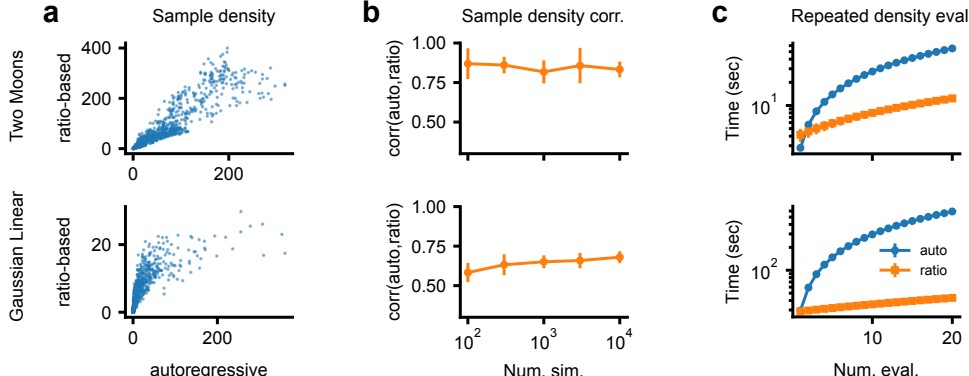

Figure D-6: **Comparison of density evaluation approaches on Two Moons and Gaussian Linear task. (a)** Correlation between density estimates from the autoregressive and ratio-based approaches for two observations using $10^4$ simulations. **(b)** Pearson correlation between the two density estimates across varying simulation budgets for both tasks. **(c)** Evaluation time (log scale) for an increasing number of density evaluations using an Nvidia H100 GPU, showing the crossover point where the ratio-based approach becomes more efficient.

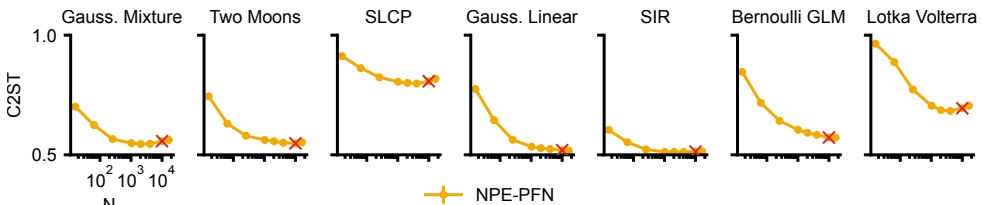

Figure D-7: **SBI benchmark results with varying number of filtered simulations.** C2ST performance of filtered NPE-PFN as a function of the number of filtered simulations $N_{\text{filter}}$. For all tasks, a simulation budget of $10^5$ is used. The red cross marks the default choice of $N_{\text{filter}} = 10^4$.

We observe a strong correlation between autoregressive and ratio-based density evaluations for the Two Moons task and a somewhat weaker correlation for the higher-dimensional Gaussian Linear task (Fig. D-6a for a simulation budget of $10^4$). This trend persists across simulation budgets (Fig. D-6b). In terms of runtime, the ratio-based method has an initial cost due to posterior sampling, making it slower for the first evaluation. However, once samples are obtained, subsequent evaluations are significantly faster—an advantage that becomes very noticeable as the number of density evaluations increases (Fig. D-6c; note the logarithmic y-axis).

Therefore, in scenarios that require repeated density evaluations for a single observation—such as rejection sampling in TSNPE-PFN—the ratio-based approach offers a highly favorable trade-off: It provides sufficient accuracy at a fraction of the computational cost.

### D.6 Varying the number of filtered simulations

When filtering simulations, we always make full use of TabPFN's recommended maximal context size of $10^5$ data points. However, in principle, the number of simulations $N_{\text{filter}}$ selected by the filter is a hyperparameter that can be optimized. Here, we investigate the impact of this hyperparameter by evaluating the performance of NPE-PFN on the SBI benchmark tasks and a simulation budget of $10^5$. Specifically, we vary the number of filtered simulations $N_{\text{filter}}$, setting it to 16, 64, 256, 1024, 2048, 4096, and 16384, and compare the resulting performance to our default choice of $10^4$.

For small values of $N_{\text{filter}}$, the performance of NPE-PFN deteriorates across all tasks (Fig. D-7). In contrast, for $N_{\text{filter}} = 1024$ or larger, performance is very similar to that of our default choice with $10^5$ simulations. For some tasks, a slight decrease in performance is observed for the largest value of $N_{\text{filter}}$, likely because this exceeds the recommended context size of TabPFN. Importantly, our default choice is always optimal or near-optimal. These results suggest that the recommended maximal context should be utilized fully. Nevertheless, these results also indicate that good performance can be achieved with smaller filter sizes (e.g., $N_{\text{filter}} = 2048$) to reduce the computational load.

### D.7 Varying feature and noise distributions

Here, we investigate the robustness of NPE-PFN to different types of noise. To evaluate the robustness in such cases, we construct a variant of the Gaussian linear task from Lueckmann et al. [27] with non-Gaussian features and noise

$$\boldsymbol{\theta} \sim p_{\text{feat}}(\boldsymbol{\theta}), \tag{D-8}$$

$$\boldsymbol{x} \sim \theta + p_{\text{noise}}(\boldsymbol{x} \mid \boldsymbol{\theta}). \tag{D-9}$$

Both $p_{\text{feat}}(\boldsymbol{\theta})$ and $p_{\text{noise}}(\boldsymbol{x} \mid \boldsymbol{\theta})$ are chosen from a set of distributions with varying support and tail behavior:

- Cauchy distribution $\text{Cauchy}(0, s)$,
- Laplace distribution $\text{Laplace}(0, s)$,
- Logitnormal distribution $\sigma(\mathcal{N}(0, s^2))$ with $\sigma(x) = \frac{1}{1+e^{-x}}$,
- Normal distribution $\mathcal{N}(0, s^2)$,
- Student's t-distribution $t_5 \cdot s$ where $t_5$ represents a Student's t-distribution with 5 degrees of freedom,

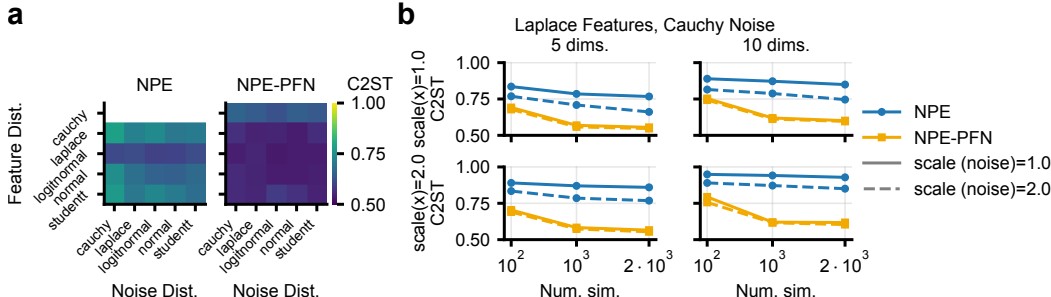

Figure D-8: **Robustness to feature and noise distributions.** (a) Heatmaps showing C2ST performance for all combinations of feature and noise distributions with $\dim(\boldsymbol{\theta}) = 5$, $s = 1.0$, and $10^3$ training simulations. White cells indicate failed training. (b) C2ST across training set sizes for an example combination of distributions. These panels are zoom-ins from (a), highlighting the stability of NPE-PFN under varying distributional assumptions.

where $s$ is the scale parameter. We vary the feature dimension, the scale of feature and noise distributions, and the number of training simulations, and compare the performance of NPE-PFN against NPE.

We first evaluate performance via C2ST across all combinations of feature and noise distributions for $s_{\text{features}} = s_{\text{noise}} = 1.0$, $\dim(\boldsymbol{\theta}) = 5$, and $10^3$ simulations. The heatmap shows that NPE-PFN outperforms NPE consistently, with while cells indicating training failure primarily for NPE (Fig. D-8a). For the example of Laplace feature and Cauchy noise distributions, we provide results as a function of simulation budget, confirming the robustness of NPE-PFN across distributional shifts (Fig. D-8b). Notably, while NPE achieves performance drops on several non-Gaussian configurations, NPE-PFN maintains stable performance.

These results suggest that, although TabPFN is pre-trained using priors based on structural causal models (SCMs) with uniform or Gaussian root noise, nonlinear transformations in the SCM induce rich marginal distributions and allow generalization to a wide range of distributions. NPE-PFN inherits this robustness, and its performance is largely invariant to different feature or noise distributions. Failures of NPE, especially with Cauchy features, are caused by instability in z-scoring, which NPE-PFN avoids through default preprocessing such as the Yeo–Johnson transform [96]. Thus, standard NPE performance could probably be improved by applying appropriate transformations. Because NPE-PFN takes advantage of the automatic preprocessing performed within TabPFN, it provides reliable performance without manual preprocessing, making it a robust and user-friendly solution in practical settings.

### D.8 Order of autoregressive sampling

To sample from multi-dimensional (conditional) distributions with TabPFN, we use it in an autoregressive manner. An important question is whether the order in which we sample the dimensions matters for, e.g., the quality of the inferred posterior distributions. To investigate this question, we rerun NPE-PFN on the benchmark tasks from Sec. 3.1, but permute the order in which we sample the parameter dimensions. Two of the seven benchmark tasks are permutation invariant by design (Gaussian Mixture and Gaussian Linear), and we replace these with tasks where a distribution is constructed autoregressively. The first is a simple nonlinear task given by

$$[x_1, x_2] \sim \mathcal{N}(0, 1), \tag{D-10}$$
$$y_1 \sim \mathcal{N}(x_1, 1), \tag{D-11}$$
$$y_2 \mid y_1 \sim \mathcal{N}(\sin(y_1 + x_2), 1), \tag{D-12}$$
$$y_3 \mid y_1, y_2 \sim \mathcal{N}(y_2^2 + y_1, 1), \tag{D-13}$$
$$y_4 \mid y_1, y_2, y_3 \sim \mathcal{N}(y_1 y_2 + y_3, 1). \tag{D-14}$$

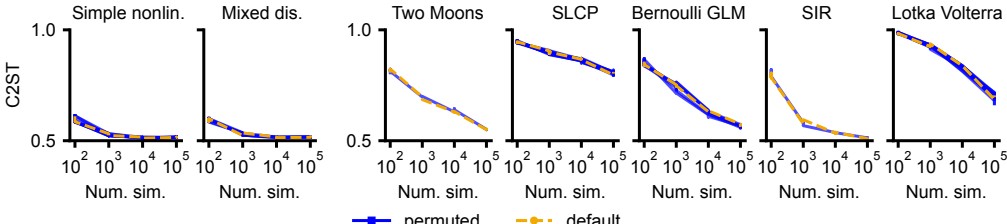

Figure D-9: **Autoregressive ordering.** C2ST across varying simulation budgets for two additional synthetic tasks, and five tasks from the SBI benchmark that are not permutation invariant. Blue lines indicate the average C2ST accuracy over three seeds for up to ten (random) permutations of the sampling order. Dashed orange lines indicate the default order.

The second is a mixed distribution task, given by

$$x = [x_1, x_2] \sim U(-2, 2), \tag{D-15}$$
$$y_1 \sim \mathrm{Gamma}(\mathrm{shape} = 1 + |x_1|, \mathrm{scale} = 1), \tag{D-16}$$
$$y_2 \mid y_1 \sim \mathrm{Uniform}(0, y_1 \cdot 2 + |x_2|), \tag{D-17}$$
$$y_3 \mid y_1, y_2 \sim \mathrm{Beta}(\alpha = 1 + y_1, \beta = 2 + y_2). \tag{D-18}$$

As in Sec. 3.1, we compute the C2ST with respect to the ground truth for evaluation. We run NPE-PFN for the default autoregressive order (as defined in the SBI benchmark or in the equations above) and up to ten (random) permutations over three random seeds. Note that tasks with two or three dimensions have only two or six possible permutations, respectively (including the default one), so random subsampling is not required.

Across all benchmark tasks and simulation budgets, NPE-PFN achieves nearly identical performance across the default and permuted orders (Fig. D-9). That is, for these benchmark tasks, the performance of NPE-PFN is not affected by permuting dimensions. These results show that NPE-PFN is not sensitive to the order in which the parameters are sampled. As a result, users need not worry about providing NPE-PFN with an "optimal" order. While differences in performance may be possible for very high dimensions or artificial examples, for the applications we consider here, an arbitrary permutation is sufficient.

## D.9  Unconditional density estimation on the UCI datasets

Here, we apply TabPFN on some classical *unconditional* density estimation benchmark tasks from the UCI repository [56] as used in several other works [41, 42, 97]. Note that while we still perform density estimation here, it does not involve posterior distributions. Thus, we do not refer to this approach as NPE-PFN. Specifically, we consider the Gas, Power, Hepmass, and Miniboone datasets. These tabular datasets range in dimensionality from 6 to 43 features and contain between $3.1 \cdot 10^4$ and over 1 million samples. We here investigate the unconditional density estimation performance of TabPFN in the low sample regime in comparison to a neural spline flow (NSF) [42]. As in previous works, we compute the negative log-likelihood (NLL) under a held-out test set. While we evaluate on the full test sets, we only use $10^3$, $10^4$, or (if applicable) $10^5$ samples for training. For the two larger settings, we use partitioning (Sec. B.4) with 10 clusters when estimating densities with TabPFN.

For $10^2$ and $10^3$ samples, TabPFN achieves a smaller negative log-likelihood on all four datasets (Fig. D-10). Similarly, for the $10^4$ samples, TabPFN outperforms NSF on all but the power dataset. Interestingly, for the lower dimensional datasets, clustering further reduces the NLL despite having access to the same total number of samples. On the other hand, for the higher dimensional datasets, clustering increases the NLL. For $10^5$ samples (or $3.1 \cdot 10^4$ samples for the Miniboone dataset), clustering improves the NLL over any $10^4$ sample setting because it allows TabPFN to access a larger number of samples. However, NSF performs better than or equal to TabPFN on all datasets except the Gas dataset at $10^5$ samples.

These results suggest that TabPFN is a capable (unconditional) density estimator, especially in the low sample regime.

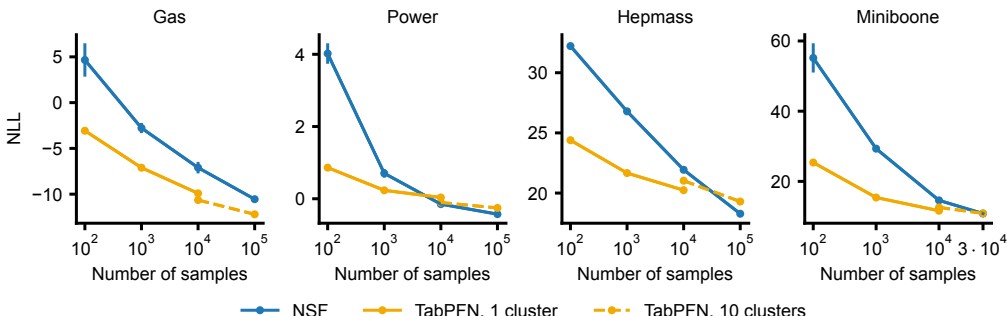

Figure D-10: **Results for unconditional density estimation on the UCI datasets.** Negative-log-likelihood (NLL) for TabPFN (with 1 and 10 clusters) and the neural spline flow (NSF) across the different UCI datasets. Dots indicate averages, and bars show standard deviation over five independent runs.

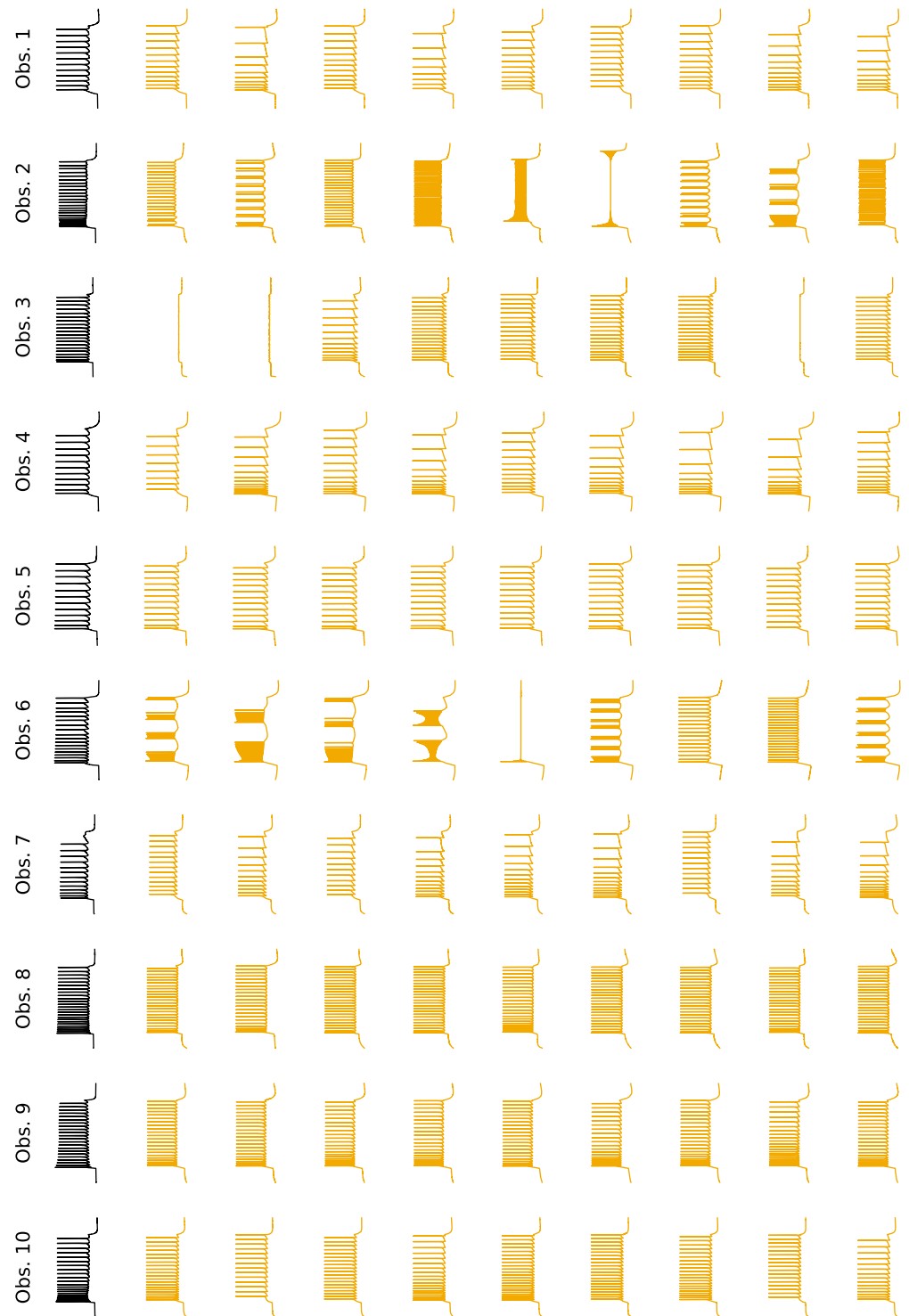

Figure D-11: **Posterior predictives of TSNPE-PFN** for real observations from the Allen cell type database and a simulation budget of $10^4$. Note that the inference is not performed directly on the action potential time series but on seven summary statistics computed from it.

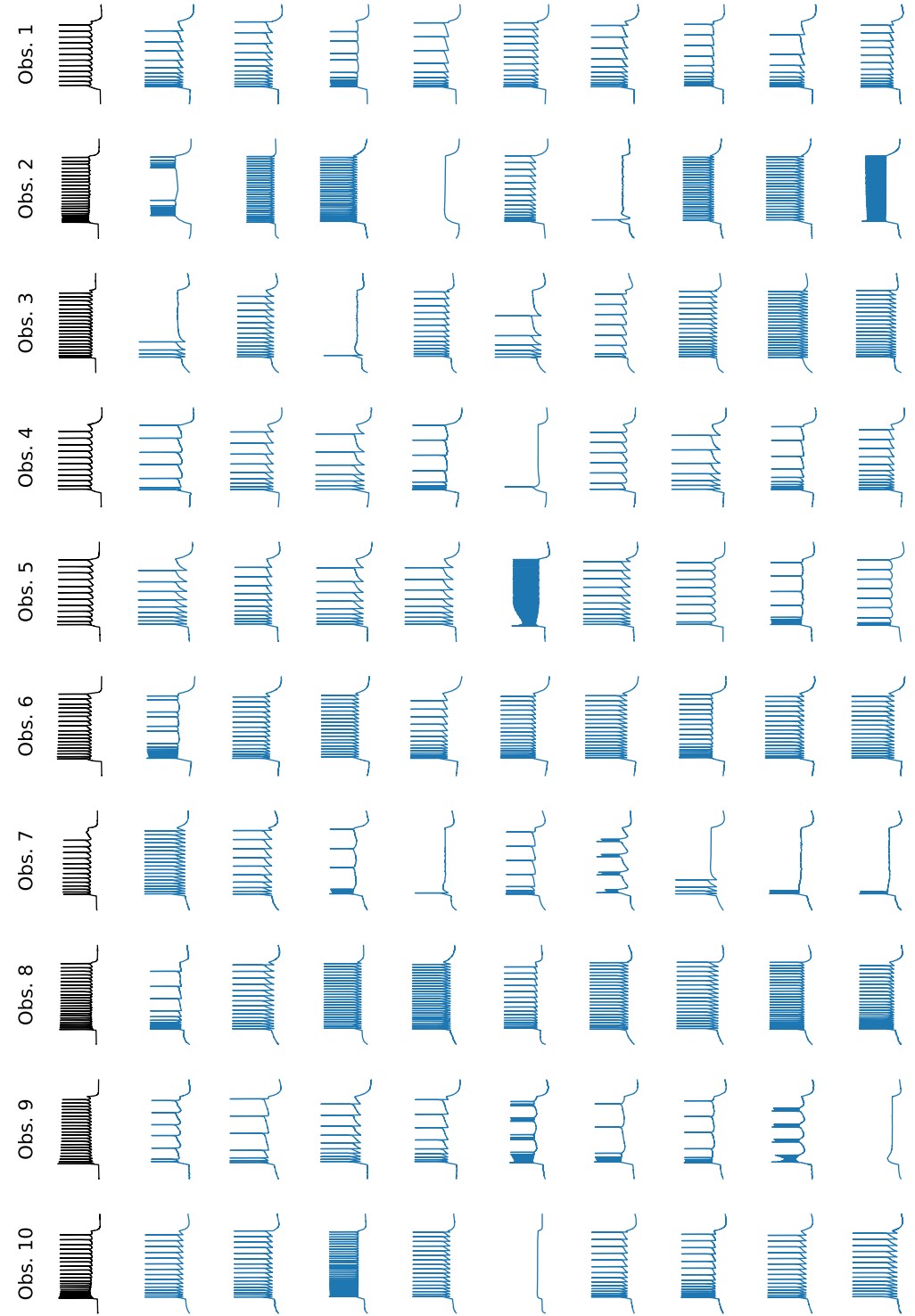

Figure D-12: **Posterior predictives of the TSNPE baseline** for real observations from the Allen cell type database and a simulation budget of $10^4$. Note that the inference is not performed directly on the action potential time series but on seven summary statistics computed from it.

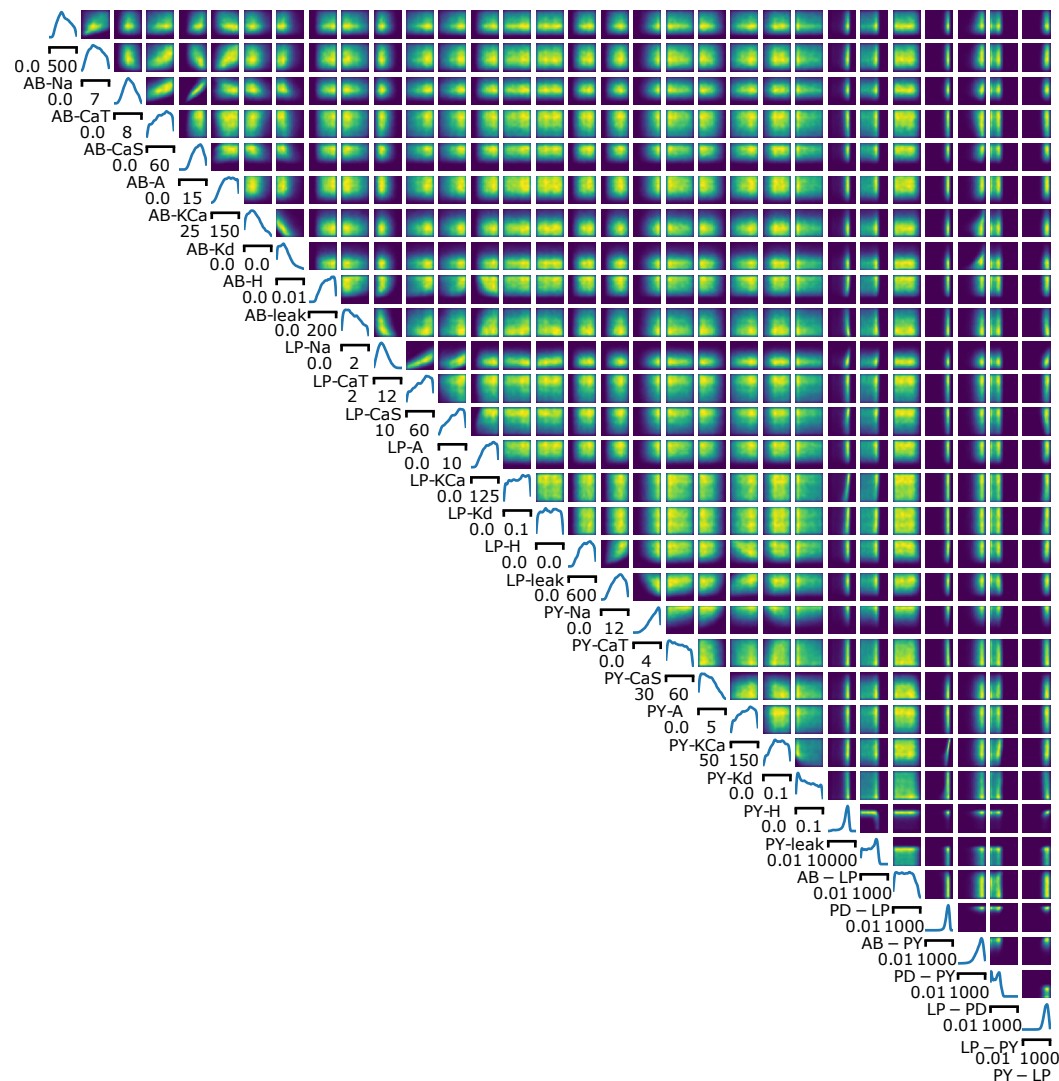

Figure D-13: **Pyloric simulator.** Posterior distributions for all 31 parameters estimated by TSNPE-PFN.

