# OpenReview forum: "Effortless, Simulation-Efficient Bayesian Inference using Tabular Foundation Models"
_NeurIPS.cc/2025/Conference — NeurIPS 2025 poster_

### Official Review · Reviewer_RF8X · 2025-06-19

**Clarity:** 4
**Significance:** 3
**Originality:** 4
**Rating:** 5
**Confidence:** 3

**Summary:**

This work tackles an idea that has been gaining more and more traction in the AI4Science literature : how to leverage foundational models to accelerate science discovery? The authors use the recently proposed TabPFN architecture, initially conceived for univariate regression/classification problems, in a simulation-based inference setting. The main idea is to recast TabPFN as a univariate density estimator and to plug into an autoregressive factorisation of a possibly multivariate posterior distribution.

**Questions:**

1) I would have appreciated having a clearer explanation (maybe with a figure? diagram?) of how the sampling of the final posterior approximator is done. The approach is a bit convoluted envolving a classifier, TSNPE, TabPFN, rejection sampling, etc. It took me a bit of time to understand how each brick connects and I'm not sure that every reader would have the energy/time to do the same.

2) In the autoregressive factorization, how does the ordering of the choice of dimensions play a role? Is there an optimal way in practice? Or is the TabPFN model completely equivariant to the ordering of the input dimensions?

**Ethical Concerns:**

["NO or VERY MINOR ethics concerns only"]

**Final Justification:**

I believe the paper has its merits for an acceptance. I don't think is worth a spotlight or something like that, but the authors did a good rebuttal and would not be against seeing their work at the conference. I can't, however, justify increasing my already rather positive score.

**Limitations:**

Yes

**Quality:**

3

**Strengths And Weaknesses:**

The paper reads very well and is based on a rather popular and easily understandable engine, i.e. TabPFN. The way I see, the main methodological contributions of the authors were:
- Recasting TabPFN to SBI by leveraging the autoregressive factorisation of pdfs
- Despite the rather tedious and potentially slow sampling that entails from the pdf factorisation, the authors proposed a nice "ratio trick" to enable faster sampling from the trained approximate posterior at inference time

---

> ### Author Rebuttal · Authors · 2025-07-29
>
> We thank the reviewer for their overall positive evaluation of our manuscript – finding that our “paper reads very well” and acknowledging that the use of foundation models in SBI “tackles an idea that has been gaining more and more traction”.
>
> **Q1. Clear explanation of final posterior sampling.**
>
> We appreciate the suggestion and will provide a more thorough explanation in the updated version. In particular, we will add detailed pseudocode to the appendix to provide a clear and rigorous overview of our TSNPE-PF approach.
>
> **Q2. Optimal autoregressive factorization.**
>
> We discuss the question of whether an optimal order of the parameter dimension for TabPFN exists in the Appendix Section. D.6, Fig. D-7. Indeed, TabPFN is not equivariant to the ordering of the inputs. In general, given a different ordering, each one-dimensional conditional distribution, which TabPFN has to predict, changes. We conduct experiments for two synthetic tasks and five benchmark tasks, which are not permutation equivariant. Specifically, we randomly permute the parameter dimensions and evaluate the performance. Our results indicate that NPE-PF is relatively insensitive to the order in which parameter dimensions are permuted.
>
> We additionally tested this on our 31‑dimensional pyloric network example: Here the total number of possible permutations is greater than $10^{33}$.  Finding the optimal permutation in this space is highly challenging. To gain at least some insight, we performed an ablation study on 50 distinct random permutations using the context set identified by the final‑round posterior. We compute the presented metrics with 1000 simulations. We found that the energy score varies between 0.081 - 0.094 and the valid rates between 96% - 98% (5% and 95% quantile each). This is in line with what we found for the default order reported in the paper, Appendix D.6, Fig. D-7. However, these results do not prove the absence of an optimal permutation that would significantly improve performance. Finding such a permutation is likely to be very challenging in general and would present an interesting opportunity for future research.

---

> > ### Comment · Reviewer_RF8X · 2025-08-08
> >
> > I appreciate the authors response and their answers to my questions. I will keep my positive score.

---

### Official Review · Reviewer_BTVU · 2025-06-26

**Clarity:** 4
**Significance:** 3
**Originality:** 4
**Rating:** 5
**Confidence:** 4

**Summary:**

This paper proposes using Tabular Foundation Models, specifically TabFPN for SBI. The new approach is evaluated on a few standard benchmarks and two more challenging models.

**Questions:**

- What is the inference time of the methods in the two real world case studies 3.3 and 3.4?
- How fast is density evaluations of NPE-PF relative to NPE with normalizing flows?
- Since we are using TabPFN in an autoregressive manner, is the number of features (only) d_x or d_x + d_theta(<j)? Since we seem to use both data and previous parameters as features? Perhaps you say it somewhere but it wasn’t entirely clear to me when reading the paper.
- How good are the sbi pacakge defaults for the flows? Could it be that NPE is just less good the NPE-PF because the defaults are far from ideal? How much more efficient can things be if we tune the architectures to the given examples? I know this requires effort and expertise, but if we really case about simulation-budget, than this optimizing is probably the smallest evil. I am just worried that the better NPE-PF results are partially due to bad NPE defaults.
- When indexing the pre-conditioning/training data, you sometimes use M and N. What is the difference between M and N?
- In 2.2 You say, “predict the next dimension of data”. Do you mean “predict the next dimension of parameters”?
- In the calibration results Figure D-3, could you include the confidence envelope assuming well calibration? And perhaps consider difference plots so one has an easier time judging (mis)-calibration which is currently very hard to see in the existing plot.
- In MMS case study 3.2, I am not sure I understand what you considered the “reference”. Is it the analytical posterior of the misspecified model? Or is it some other target/reference? The text appears unclear in that regard.
- In the limitations, you speak of “continuous parameters”. Since the benchmarks you use all have continuous parameters, I don’t understand what you mean by that term as a limitation?

**Ethical Concerns:**

["NO or VERY MINOR ethics concerns only"]

**Final Justification:**

Authors rebuttal addressed most of my concerns.

**Limitations:**

A lot of limitations are clearly discussed although the magnitude of these limitations may be understated to some degree. See "weaknesses".

**Quality:**

3

**Strengths And Weaknesses:**

Strengths:
- The idea is very cool!
- The reported results are largely convincing, especially when it comes to simulation-efficiency.

Weaknesses:

- The used standard (low dimensional) benchmarks are a bit outdated and do not resemble many challenges we face in practice. I know they continue to be reused (because they are easy to fit well I assume) but I don’t see how results are of much use to judge the real-world applicability of the benchmarked methods. In addition, the evaluation of only 10 test data sets is a limitation especially since we apply amortized methods, where we should easily validate results on 100s of datasets.
- Some of the results are presented in a way that it boarderlines on selective reporting/discussing. For example, the sampling times of the new method NPE-PF seems to be around 10-100 seconds per dataset (based on what I see in Figure 2). In contrast, NPE with normalizing flows requires only 0.1 seconds (or less?) per dataset. This is shown (in a very small manner) in Figure 2, the text says NPE-PF is slower than NPE. While true, the multiple orders of magnitude in speed difference are never explicitly mentioned. I suspect they could easily be missed by the unattentive reader. If we combined the slower training time of NPE with the faster inference time, when would NPE become overall faster again than NPE-PF (given similar calibration)?
- The fact that a summary (embedding) network isn’t naturally fitting into the framework is another limtation of the approach. For almost all non-trivial data, we would want a summary network to handle the data structure properly. Yet, to use it in combination with NPE-PF, we have to pre-train it separately, reducing the argument of NPE-PF being “training-free”. How much additional training and simulation-budget do we need in this case? Is it substantially smaller (or faster) than training NPE+summary network combined? Also, I fail to find information in B.2 how good the “sufficient statistics” approximation described in EQ B-3 is. What are potential issues with this approach?

---

> ### Author Rebuttal · Authors · 2025-07-29
>
> We thank the reviewer for their overall positive evaluation ("very cool idea," "largely convincing") and relevant suggestions that led us to add new tasks, a full cost analysis, and an NPE hyper-parameter sweep, which significantly strengthened our work.
>
> We first address the main issues and discuss new results (**A1–A3**), then give detailed replies to questions (**Q1–Q9**).
>
> - **Benchmark limitations:** We use an established benchmark for SBI, but now add new tasks and metrics with more test points for broader coverage (**A1**).
> - **Computational cost:** We acknowledge that our discussion was imprecise. We will update the manuscript to clearly write that the inference time of NPE-PF is orders of magnitude slower than that of NPE, and will add a careful discussion of our cost analysis (**A2**).
> - **Embedding nets:** We do not claim NPE-PF outperforms NPE when embedding networks are needed; we merely explore this practical/frequent use-case (**A3**).
>
> **A1. Additional tasks and datasets.**
>
> We agree with the reviewer's assessment that the "used standard (low-dimensional) benchmarks" with "only 10 test data sets" have limitations. However, as the reviewer notes, this SBI benchmark is still widely used and tasks are not "easy to fit" for NPE methods (e.g., SLCP or Lotka-Volterra). Nonetheless, we extended our current evaluation suite with several new tasks from other SBI papers [1,2], also including particularly computationally demanding physics simulations ("streams", 30 min. per simulation, 2 dim parameters, 199 dim data). For the new tasks, we compute the average negative log likelihood (NLL) on 1000 test points. If the tasks provide a ground truth posterior, we use 100 test points in the C2ST evaluation.
>
> On these new tasks, NPE-PF again shows strong performance, especially for the smaller simulation budgets, where it is superior to NPE for all tasks. For the detailed results, we refer to the response to reviewer Vbbm, Table V1/V2.
>
> **A2. Analysis of computational cost.**
>
> We ran additional timing experiments. As expected, inference time mostly scales with an increasing parameter dimension and a larger number of simulations. See Table w2 in the response to reviewer wFAY for details.
>
> Providing a general answer to the question of when "would NPE become overall faster again" is hard because it is task and application dependent. Disregarding simulation time, NPE-PF quickly becomes less efficient than NPE: For example, assuming 100000 available simulations (8 dim parameters, 8 dim data), NPE takes around 60–90 minutes to train. After training, inference will be effectively instantaneous. In contrast, generating 10000 posterior samples for a single observation using NPE-PF takes ca. 50 seconds (Table w2, response wFAY). Therefore, the training time of NPE amortises after evaluating around 90 observations. However, for expensive simulators (e.g. stellar streams), simulation becomes the bottleneck and the simulation efficiency of NPE-PF compensates for the slower inference time. For sequential inference, the situation favours TSNPE-PF, because everything needs to be rerun per single observation.
>
> We will add a thorough discussion of the computational times to the manuscript. Our goal with NPE-PF is not to replace NPE, but to provide an easy-to-use and simulation-efficient alternative, highlighting strengths and limitations such that users can make an informed choice.
>
> **A3. NPE-PF with summary (embedding) networks.**
>
> We thank the reviewer for raising this point and agree that it requires clarification:
>
> Our main goal with NPE-PF is to introduce an easy-to-use method that allows users to obtain good results with few simulations, given a sufficiently low-dimensional inference problem (<500dim). In high-dimensional cases that need an embedding net, the number of required simulations typically also grows large, and NPE is expected to be superior. Thus, NPE (with embedding net) and NPE-PF serve different use cases.
>
> To further clarify, we do not claim that NPE-PF will be superior to NPE when embedding nets are needed, nor that our approach for pretraining them is the best. Our goal was simply to investigate how NPE-PF performs together with an embedding network or generally a neural representation of data. For more common datatypes, networks to compress data can be already available. The question of how much "additional training and simulation budget" is needed to obtain "good" neural representations is independent of NPE-PF and beyond the scope of this work. For related questions including "how good the 'sufficient statistics' approximation" is, see [3,4].
>
> Finally, even when simulators are high-dimensional, a set of well-engineered, low-dimensional summary statistics is still routinely used in many SBI applications. Such cases include expensive simulators, where training an embedding net is infeasible, or misspecified simulators, where summary statistics provide increased robustness. In these cases, NPE-PF provides a powerful alternative to traditional approaches, even in high-dimensional settings.
>
> **Q1. Inference time of real-world tasks.**
>
> For our experiment on the recordings from the Allen cell types database (Sec. 3.3), the inference times are as follows:
> - TSNPE-PF, 1000 sim.: ca. 10 min, 10000 sim.: ca. 60 min.
> - TSNPE, 1000 sim.: ca. 4 min, 10000 sim.: ca. 40 min.
>
> Compared to TSNPE with 10000 simulations, our approach achieves comparable inference quality using only 10% of the budget.
>
> For the Pyloric simulator (Sec. 3.4), inference takes 2-7 minutes per round (including simulation). See Appendix C.4 for an extended discussion and inference time comparison to baseline methods. Note that SNVI baseline needs 30 hours runtime on CPU, whereas the SUNLE baseline requires 3 hours on GPU to acquire a limited set of samples (excluding the time to simulate the much larger initial dataset).
>
> **Q2. Inference time of density evaluation.**
>
> Autoregressive density evaluation is as fast as sampling. For the exact timings, please refer to Table w2, wFAY. The ratio-based density evaluation is much faster than the autoregressive one. See a detailed analysis in Appendix D.4, Fig. D-5c. Both ways of evaluating densities with TabPFN are slower than a flow and we will make sure to state this more clearly in the manuscript.
>
> **Q4. Sweep for hyperparameters.**
>
> We agree that the performance of NPE on individual tasks can likely be increased, provided users have enough expertise and computational resources. Therefore, we have conducted an extensive computational experiment, where flow and optimization hyperparameters are swept for each task and simulation budget. Sweeping results are in Table B1 and details in caption. Even for the smaller budgets, this procedure required a substantial amount of compute, taking several hours per setting.
>
> Sweeps give only modest gains—mainly for small budgets where exhaustive search is feasible—and demand far more time or better search for larger budgets. Tuned NPE still trails NPE‑PF on most low‑budget tasks. Importantly, NPE‑PF’s edge is its strong default performance: a better “perfect NPE network" certainly exists, but finding it with few simulations is challenging or basically impossible, and not "effortless".
>
> Finally, this strong default performance becomes crucial for sequential inference, where sweeping is considerably more challenging and costly (long inference times, changing training conditions over rounds, difficulty of selecting appropriate metrics to optimize).
>
> **Table B1 Sweep results.** C2ST averages across 5 seeds (SDs negligible, space constraints). Search space: flow type, number of transforms, hidden layer size, learning rate, and batch size. We use random search to explore over 1000 configurations with respect to the best validation loss. We limited the maximum time per setting to 10 hours.
> |Method|Gaussian Mixture|Two Moons|SLCP|Gaussian Linear|SIR|Bernoulli GLM|Lotka Volterra|
> |-|-|-|-|-|-|-|-|
> |**100sim.**||||||||
> |NPE-PF|**0.72**|**0.81**|**0.95**|**0.64**|**0.78**|**0.85**|**0.98**|
> |NPE-Sweeper|0.79|0.89|0.98|0.87|0.92|0.97|1.00|
> |**1000sim.**||||||||
> |NPE-PF|0.72|0.69|**0.90**|**0.55**|**0.58**|**0.74**|**0.93**|
> |NPE-Sweeper|**0.65**|**0.68**|0.94|0.66|0.72|0.80|0.99|
> |**10000sim.**||||||||
> |NPE-PF|0.68|0.63|**0.86**|**0.52**|**0.54**|**0.63**|**0.84**|
> |NPE-Sweeper|**0.54**|**0.57**|**0.86**|0.57|0.55|0.66|0.96|
> |**100000sim.**||||||||
> |NPE-PF|0.56|0.55|0.81|**0.52**|**0.51**|**0.57**|**0.69**|
> |NPE-Sweeper|**0.52**|**0.53**|**0.71**|0.53|0.52|0.58|0.88|
>
> **Q3, 5-9. Clarifications.**
>
> - **Number of features.**
> Indeed, the dimensionality supported by NPE-PF is the combined dimensionality of observation and parameter space.
>
> - **Calibration results.**
> We agree that using difference plots and including the confidence envelope will increase readability, and will update the plot accordingly.
>
> - **Misspecification.**
> For the misspecification experiment, C2ST are computed with respect to non-misspecified ground truth posterior, as in [5].
>
> - **Other phrasing.**
>   - With "continuous parameters", we meant infinite-dimensional parameter spaces such as spatially varying parameters.
>   - Indeed, we meant "parameters" instead of "data". Thanks for catching this.
>   - With M we refer to the size of the datasets used for pretraining TabPFN, and with N we denote the number of simulations to be used with NPE-PF.
>
> We will correct/phrase the above points more clearly.
>
> [1] Hermans et al. "A trust crisis in simulation-based inference? Your posterior approximations can be unfaithful." 2021.
> [2] Gloeckler et al. All-in-one simulation-based inference. 2024.
> [3] Yanzhi et al. "Neural approximate sufficient statistics for implicit models.", 2020.
> [4] Yanzhi et al. "Is learning summary statistics necessary for likelihood-free inference?.", 2023.
> [5] Schmitt et al. "Detecting model misspecification in amortized Bayesian inference with neural networks: An extended investigation.", 2024.

---

### Official Review · Reviewer_wFAY · 2025-07-01

**Clarity:** 3
**Significance:** 3
**Originality:** 2
**Rating:** 4
**Confidence:** 4

**Summary:**

The paper explores the use of in-context-learning (ICL) in SBI settings. It employs TabPFN as backbone model, avoiding the use of inference networks, for instance, in NPE (neural posterior estimation). The proposal employs a sequential inference step to handle models with multiple parameters and adopts nearest-neighbor-based filtering to overcome the limited context size of TabPFN. Experiments on synthetic and real data demonstrate that the proposal is more sample-efficient (in terms of number of simulations) and robust to model misspecification than baseline frameworks (e.g., NPE, NLE).

**Questions:**

1. Could the authors elaborate on the sensitivity of the performance to some modeling choices, e.g., number of neighbors?
2. What is the computational impact of the proposal? How much does it improve over standard NPE, for instance?
3. On which datasets is TabPFN pre-trained? Could these datasets say something about settings (problems) in which the proposal work?
4. What makes the TabPFN model the best foundation model for SBI? For instance, how does the method in [1] perform?

Please, see "Weaknesses" for additional questions.

[1] TabICL: A tabular foundation model for in-context learning on large data?, ICML 2025.

**Ethical Concerns:**

["NO or VERY MINOR ethics concerns only"]

**Final Justification:**

The paper tackles a relevant problem (sample efficiency) in SBI using foundational models. The empirical results are promising and significant. During the rebuttal, the authors provided clarifications regarding TabPFN and presented additional results that alleviated some of my initial concerns. Overall, I am leaning towards acceptance.

**Limitations:**

The paper includes a paragraph "Limitations". However, I believe the paper should better explore the limits of using TabPFN (or ICL in general) for SBI (see Weaknesses).

**Paper Formatting Concerns:**

The paper seems to follow the formatting guidelines.

**Quality:**

2

**Strengths And Weaknesses:**

**Strengths**
- Overall, the empirical results are promising and significant.
- This is the first paper to explore ICL approaches in the context of SBI.
- The paper is well-written and relatively easy to follow.


**Weaknesses**
- The paper does not provide in-depth insights into when and why the ICL approach works across different tasks or simulators, limiting the contribution to the community. Is TabPFN expected to beat NPE in all problems? What are the limits of TabPFN? Is there any relation between the datasets on which TabPFN was trained and the SBI tasks? The paper briefly mentions "modeling complex parameter spaces—such as spatially varying or continuous parameters—remains challenging". Where does this conclusion come from?

- The paper aims for an "effortless" SBI approach. However, the paper does not mention how much time the proposal saves compared to the baselines. I believe the paper lacks a comparative study from a computational (time) perspective. In practice, how many hours would a practitioner save by using the proposed approach?

- The paper would also benefit from an "ablation-like" study. For instance, how does the performance change as a function of the number of neighbors N? How many samples are filtered (on average)? This could provide more insights into how TabPFN works and its limits.

- From a technical perspective, the paper brings little to the table. In particular, the application of the ICL approach is straightforward, and the solution used to overcome the limited context size  (i.e., nearest-neighbor-based filtering) is not novel.

---

> ### Author Rebuttal · Authors · 2025-07-29
>
> We thank the reviewer for acknowledging that “the paper is well-written”, and that “the empirical results are promising and significant”. We also appreciate the recognition that ours is the “first paper to explore ICL approaches in the context of SBI”.
>
> In response to the reviewer’s suggestions, we have:
> - Conducted experiments across a broader range of filter sizes (Table w1).
> - Added a rigorous computational time comparison (Table w2).
> - Introduced several new, more challenging inference tasks to cover a larger range of tasks.
>
> Below, we first address the reviewer’s main concerns and then provide detailed answers to their specific questions (**Q1–Q4**).
>
> - **Technical novelty of contribution:**  Our paper tackles two central challenges in SBI: the high simulation cost of traditional methods and the expertise required to train conditional density estimators. Our method, NPE-PF, provides a novel and elegant way to alleviate both of these issues. Furthermore, TSNPE-PF uses new ideas for fast in-context density estimation with classifier-based density ratios, and shows strong performance in real-world inference tasks (Fig. 4/5). To our knowledge, our paper hence introduces novel methods to the SBI literature that address problems relevant to the community, regardless of any perceived lack of technical complexity. Instead, we view this simplicity as a strength, lowering the barrier for adoption by researchers and practitioners.
> - **No in-depth insights into when and why ICL works:** The reviewer states that our work “does not provide in-depth insights into when and why the ICL approach works” and asks “Is TabPFN expected to beat NPE in all problems?” While we did not test on *all* problems, we did test on a considerable number of problems and now further increased this number (results in Table V1/V2 in response to Reviewer Vbbm). All these results support our main claim that NPE-PF outperforms NPE on small simulation budgets, and is competitive for large ones. Yet, we agree that the manuscript could be improved by providing a more elaborate interpretation and conclusion of these results to better communicate the insights into when and where ICL likely works well (see **Q1-Q4** for details).
>
>
>
> **Q1. Ablation on number of neighbours for filter.**
>
> We thank the reviewer for their suggestion. We have conducted an experiment on all SBI benchmark tasks using 100k simulations and varying filter sizes (16, 64, 256, 1024, 2048, 4096, 16384) in comparison to our default choice of 10000, which is based on the TabPFN recommended context limit. Results are in Table w1.
>
>
> **Table w1: Ablation on filter size.** Average C2ST for each task (across five seeds) when varying the filter size. Standard deviations across seeds are all below 0.01, and are not reported individually. Each on a dataset of 100 000 simulations.
>
> | Filter Size | Lotka Volterra | Gaussian Mixture | Two Moons | SLCP | Gaussian Linear | SIR | Bernoulli GLM |
> |-------------|----------------|------------------|-----------|------|------------------|-----|----------------|
> | 16 | 0.96 | 0.70 | 0.74 | 0.91 | 0.77 | 0.60 | 0.85 |
> | 64 | 0.89 | 0.62 | 0.63 | 0.86 | 0.64 | 0.55 | 0.72 |
> | 256 | 0.77 | 0.57 | 0.58 | 0.82 | 0.56 | 0.52 | 0.64 |
> | 1024 | 0.71 | **0.55** | 0.56 | 0.81 | 0.53 | **0.51** | 0.60 |
> | 2048 | 0.69 | **0.55** | 0.56 | **0.80** | 0.53 | **0.51** | 0.59 |
> | 4096 | **0.68** | **0.55** | **0.55** | **0.80** | **0.52** | **0.51** | 0.58 |
> | 10000 | 0.69 | 0.56 | **0.55** | 0.81 | **0.52** | **0.51** | **0.57** |
> | 16384 | 0.71 | 0.56 | **0.55** | 0.82 | **0.52** | **0.51** | **0.57** |
>
> Across all experiments, performance decreases noticeably with small filter sizes. Large filter sizes (4096 and 16384) perform similarly to our default choice of 10,000. Importantly, our default choice is always optimal or near-optimal. These results suggest that the available context (as recommended by TabPFN) should be used fully and that little performance benefit can be gained by moving away from the default choice (see also Fig 2. “NPE-PF (unfiltered)”). Nonetheless, these results also suggest one can obtain good performance with smaller filter sizes to reduce the computational load (> 2048).
>
> **Q2. Analysis of computational time.**
>
> Beyond the timing plots in Fig 2b, we now perform additional runtime experiments (using an Nvidia A100 GPU) varying the number of simulations, and the dimensionality of parameters and simulations/observations (Table w2). As expected, the inference time mainly depends on the number of simulations in context and the parameter dimension.
>
> In general, an answer to the question “How many hours would a practitioner save?” depends on many different factors:
> 1) First, we discuss amortized inference in an example setting. With 100000 simulations (8 dim parameters, 8 dim data), NPE typically takes around 60–90 minutes to train. After training, inference will be effectively instantaneous (Fig. 2b). By contrast, generating 10000 posterior samples for a given observation using NPE-PF takes ~50 seconds (Table w2). Therefore, after sampling posteriors for around 90 observations, the training time of NPE amortises. However, this holds only assuming that one ignores the time necessary to obtain the simulations. For very expensive simulators, obtaining simulations becomes the bottleneck and the simulation efficiency of NPE-PF will more than compensate for the slower inference time. Therefore the appropriate choice of method crucially depends on the application and simulator.
> 2) Second, with sequential inference the situation changes. When using flow-based TSNPE, many networks need to be trained per observation, so fast sampling becomes less relevant. For instance, in the experiment in Section 3.4, both TSNPE and TSNPE-PF require approximately one hour with a budget of 10000 simulations (here, the simulator is fast and the simulation time is negligible). However, TSNPE-PF is much more efficient in terms of simulations, achieving similar performance with just 1000 simulations and an inference time of 10 minutes. This effect would be amplified for a more expensive simulator.
>
> We will add the timing results and above discussion to our manuscript to allow users to make an informed choice between NPE-PF and NPE.
>
>
> **Table w2: NPE-PF runtime (seconds)** for different simulation counts and parameter/data dimensionality. Each row represents varying parameter dimensions ($\theta$-dim), each column varying data dimensions (x-dim).
> ||100 sim.||||||1000 sim.||||||10000 sim.||||||
> |-|-|-|-|-|-|-|-|-|-|-|-|-|-|-|-|-|-|-|
> |$\theta$/x-dim||**2**|**4**|**8**|**16**|**32**||**2**|**4**|**8**|**16**|**32**||**2**|**4**|**8**|**16**|**32**|
> |**2**||6|4|5|6|7||7|6|5|9|8||7|8|10|14 |23|
> |**4**||11|10|10|11|13||10|11|11|13|17||15|19|24|32|48|
> |**8**||19|23|22|27|30||23|23|24|27|35||35|40|49|66|101|
> |**16**||45|47|47|53|63||49|52|54|61|77||89|99|116|152|221|
> |**32**||97|103|107|122|137||115|116|125|141|171||251|272|307|370|516|
>
>
> **Q3. TabPFN details and insights for SBI.**
>
> While NPE-PF beats NPE on all our tasks with a simulation budget of 100, it is certainly not expected that NPE-PF will be superior for all simulation budgets. Indeed, there are several benchmark tasks in which NPE performs better than NPE-PF, particularly for larger simulation budgets. In general, as the number of simulations grows large, NPE is expected to eventually outperform NPE-PF. Our main claim is that NPE-PF is simulation efficient, and performs particularly well when the number of simulations is small – a crucial and underexplored area within SBI.
>
> We agree with the reviewer that identifying the types of simulators on which NPE-PF performs well is interesting. TabPFN is trained using purely synthetic datasets generated from random structural causal models (SCMs), see [1]. In practice, we have found that NPE-PF performs particularly well on 'structured' simulators, which are similar to SCMs in that they have conditional independences. Many real-world simulators exhibit some degree of this structure (e.g. SIR, Hodgkin Huxley, etc.). Explicitly incorporating such independencies into the inference network was already shown to improve simulation efficiency [2]. We hypothesize that the pretraining on SCMs enables TabPFN to automatically detect and leverage these independencies directly from the data. A detailed investigation into these mechanisms is interesting future work, but beyond the scope of this study.
>
> In addition, we agree with the reviewer that certain formulations are misleading (i.e., “complex parameter spaces—such as spatially varying parameters” can simply be replaced with “high-dimensional”). We will make sure to increase the clarity of these statements and include the previous discussion.
>
>
> **Q4. TabICL and other foundation models.**
>
> We agree with the reviewer that using other foundation models within our approach would be highly valuable. Unfortunately, the TabICL model currently only provides in-context classification and therefore cannot be used for autoregressive sampling from the posterior, for which we use the TabPFN regression model. However, as the reviewer suggests, our proposed method is, in principle, independent of the foundation model used.
>
> [1] Hollmann et al. Accurate predictions on small data with a tabular foundation model. Nature, 2025.
> [2] Gloeckler et al. All-in-one simulation-based inference. ICML, 2024.

---

> > ### Comment · Reviewer_wFAY · 2025-08-06
> >
> > I appreciate the authors' efforts to address my concerns. While some questions regarding the underlying reasons for the success of the proposed approach remain open, I believe this is an interesting and relevant contribution to the SBI community. Thus, I have decided to increase my score.

---

> > > ### Author Response · Authors · 2025-08-07
> > >
> > > Thank you for the dedication you invested in reviewing our work, and increasing the score.
> > >
> > > We agree that a better understanding of why TabPFN works well for SBI is interesting, and should be studied in more detail in future work.

---

### Official Review · Reviewer_Vbbm · 2025-07-03

**Clarity:** 3
**Significance:** 3
**Originality:** 2
**Rating:** 5
**Confidence:** 4

**Summary:**

The paper introduces NPE-PF, a simulation-based inference method leveraging
TabPFN as a pre-trained density estimator. The use of a tabular foundation
model in the SBI pipeline enables training-free density evaluation, and builds
on useful prior knowledge from pre-training that can lessen the simulation
burden required for producing accurate posterior approximations.

**Questions:**

1. In regards to the point about inheriting TabPFN's properties: did you
   encounter any particularly surprising failure modes or poorly represented
   dynamics in your experiments with TabPFN? While TabPFN is a proven and
   generally capable model, it would be interesting to characterize when/if it
   systematically struggles under certain dynamics.
2. In Figure D-7, permutations in the dimension have little effect on the performance. This was calculated over a set of the SBI benchmarks, with a maximum of 10-dimensional parameters. Was this tested on the 31-D pyloric network example? How large of an effect might this have on even higher-dimensional inference tasks?

**Ethical Concerns:**

["NO or VERY MINOR ethics concerns only"]

**Final Justification:**

During rebuttal and discussion, the authors addressed many of my initial concerns by providing additional experimental results and discussions. This included several new inference tasks to further support the method's evaluation, along with discussions regarding its computational cost and understanding when it should be used over other methods in practice. I otherwise found the empirical results to be satisfactory, and the idea (if *somewhat* lacking in novelty) to be well-executed.

In total, I'm happy with the paper's (current or planned) experimental validation, presentation, and discussion of limitations. As a result, I maintain my initially positive inclination toward acceptance.

**Limitations:**

Yes

**Quality:**

3

**Strengths And Weaknesses:**

**Strengths**

- The paper is well-organized, includes a comprehensive literature review, and
  features clear figures and diagrams.
- Incorporating larger (foundation) models in SBI pipelines presents an
  interesting opportunity for NPE workflows, and has not been sufficiently
  explored in the literature to my knowledge. Further, while the idea of
  "swapping in" a foundation model sounds simple on its face, the methodology
  is carefully considered and attends to many important practical
  considerations in its construction (e.g., employing efficient ratio-based
  evaluation, adopting TSNPE, studying robustness to misspecification, etc).
- The reported performance evaluation is convincing, comparing the proposed
  method to relevant baselines on several common SBI tasks with clear
  advantages across many of the explored settings.

**Weaknesses**

- Provided the differences in sampling costs between NPE-PF and flow-based NPE
  methods, a more comprehensive analysis of computational costs (especially in
  high-dimensional settings) would be helpful. In particular, it would be
  practically useful to clarify the boundary beyond which one might consider a
  computationally faster method (in total), even with NPE-PF's possible sample
  efficiency gains and benefits of no training.
- The evaluation of the method on real-world environments is somewhat limited.
  While the Hodgkin-Huxley and pyloric network models are challenging tasks
  outside the SBI benchmark suite [1], evaluation on inference tasks from a
  wider range of scientific disciplines would better position NPE-PF's
  general utility.
- A principled analysis of NPE-PF's diminishing advantages over standard NPE
  methods under increased simulation burdens would be useful to include. At
  what upfront sample size is TabPFN only minimally comparatively informative
  (or even a burden)? How does this change with the dimensionality of $\theta$
  and/or $x_o$? How does a well-selected embedding network shorten the
  runway of the performance advantage? Characterizing these practical
  considerations would be useful in determining NPE-PF's feasibility for use in
  larger-scale SBI tasks.

[1] Lueckmann, J. M., Boelts, J., Greenberg, D., Goncalves, P., & Macke, J.
(2021). Benchmarking simulation-based inference. AISTATS.

---

> ### Author Rebuttal · Authors · 2025-07-29
>
> We thank the reviewer for finding our work “well‑organized,” for noting that our “methodology […] attends to many important practical considerations,” and for deeming our “performance evaluation […] convincing.”
>
> Below, we first address the reviewer’s main concerns and provide a more detailed analysis of the new results developed to address these issues (**A1**, **A2**, **A3**). We then offer detailed responses to the reviewer’s specific questions (**Q1, Q2**).
>
> - **Inference tasks on a wider range of scientific disciplines:** We agree that extending our evaluation suite to include simulators from additional fields (e.g., physics) is valuable. We have therefore added several new tasks. In **A1**, we analyze these new tasks in depth and will integrate the expanded results into the revised manuscript.
> - **Principled analysis of NPE‑PF’s diminishing advantages:** Formulating a general proposition in this regard is challenging, because it is very specific to the task and/or the application at hand. But, we believe that our (now even more) comprehensive empirical investigation of NPE-PF’s performance in various situations (tasks, dimensionalities, simulation budgets, etc.) constitutes a principled empirical examination of NPE-PF’s advantages (**A2** for details).  We will improve the discussion on this topic within the manuscript (e.g. more concrete statements, when we expect it to work well).
> - **Computational costs:**  We agree that a more precise discussion of runtime trade-offs is needed. In **A3**, we present an ablation study examining the impact of context size and data and parameter dimensionality.
>
> **A1. Additional inference tasks**
>
> The new tasks include simulators from physics (Weinberg, Stellar Streams), computer science (M/G/1 queue) [1], as well as “structured” synthetic tasks (Tree, HMM) [2]. Out of these, the stellar streams simulator stands out as particularly computationally demanding (>30 min. per simulation per CPU core; 2dim parameters, 199dim data). The tasks in [1] do not come with a ground truth posterior. We therefore evaluate them only via average log posterior density (see Table V1 for NNL metric) as well as calibration error (not shown here due to space constraints). We also extended the HMM task to 50 parameter and data dimensions as an additional test for high-dimensional, but “structured” parameter spaces (see Table V2 for C2ST metrics).
> Overall, our results for the new tasks are consistent with our previous findings. Across all tasks, NPE-PF significantly outperforms NPE at lower simulation budgets (Table V1/V2). Furthermore, NPE-PFperforms well on larger simulation budgets e.g. Tree, HMM or Streams (Table V1/V2). We will include these results in an updated version of the manuscript.
>
> **Table V1: Extended task benchmark.** Average negative log posterior density of the true parameters on 1000 test simulations. Each run was repeated five times and we report the average ± SD over runs. The better result is marked in bold.
> | Method | Weinberg | Streams | Tree | HMM | MG1 |
> |--------|----------|---------|------|-----|-----|
> | 100 sim. |
> | NPE | 0.12 ± 0.06 | 5.50 ± 0.18 | 4.58 ± 0.03 | 17.29 ± 0.11 | 3.75 ± 0.59 |
> | NPE-PF | **0.06 ± 0.01** | **3.08 ± 0.08** | **2.60 ± 0.09** | **11.44 ± 0.15** | **-0.26 ± 0.09** |
> | 1000 sim. |
> | NPE | 0.03 ± 0.06 | 3.85 ± 3.08 | 2.79 ± 2.60 | 13.63 ± 11.44 | 0.75 ± -0.26 |
> | NPE-PF | **0.02 ± 0.00** | **2.43 ± 0.09** | **2.19 ± 0.04** | **10.25 ± 0.07** | **-1.42 ± 0.08** |
> | 10000 sim. |
> | NPE | **0.01 ± 0.00** | 2.82 ± 0.08 | 2.18 ± 0.08 | 10.78 ± 0.06 | -0.60 ± 0.41 |
> | NPE-PF | 0.02 ± 0.00 | **2.21 ± 0.10** | **2.11 ± 0.04** | **10.02 ± 0.06** | **-1.88 ± 0.07** |
> | 100000 sim. |
> | NPE | **0.00 ± 0.00** | too expensive | **2.09 ± 0.07** | **9.99 ± 0.07** | -1.46 ± 0.49 |
> | NPE-PF | 0.02 ± 0.00 | too expensive | **2.09 ± 0.05** | 10.06 ± 0.04 | **-2.24 ± 0.10** |
>
>
> **Table V2: Extended C2ST evaluation.** Average C2ST (± SD over 5 runs) on new synthetic tasks using 100 random observations.
> | Method | Tree | HMM [10d] | HMM [50d] |
> |--|-|--|----|
> | 100 sims |
> | NPE | 0.87 ± 0.02 | 0.97 ± 0.00 | 1.00 ± 0.00 |
> | NPE-PF | **0.64 ± 0.01** | **0.77 ± 0.01** | **0.99 ± 0.00** |
> | 1000 sims |
> | NPE | 0.68 ± 0.02 | 0.91 ± 0.01 | 0.99 ± 0.00 |
> | NPE-PF | **0.56 ± 0.01** | **0.62 ± 0.01** | **0.77 ± 0.00** |
> | 10000 sims |
> | NPE | 0.57 ± 0.01 | 0.70 ± 0.01 | 0.97 ± 0.00 |
> | NPE-PF | **0.54 ± 0.00** | **0.56 ± 0.01** | **0.66 ± 0.00** |
> | 100000 sims |
> | NPE | **0.54 ± 0.00** | 0.59 ± 0.00 | 0.89 ± 0.00 |
> |NPE-PF|**0.54 ± 0.00**|**0.54 ± 0.00**|**0.65 ± 0.00**|
>
>
> **A2. Analysis of NPE-PF's diminishing advantages.**
>
> First, the question at which “upfront sample size is TabPFN only minimally comparatively informative (or even a burden)” is, to a certain degree, task-specific. NPE can outperform NPE-PF after $10^3$ simulations (e.g. Gauss. Mixture) or require more than $10^5$ simulations (e.g. Lotka Volterra). We evaluate NPE-PFs performance on a variety of tasks with different dimensionalities to give insights into its behaviour under various conditions.
>
> Second, whether NPE-PF becomes a burden also depends on the use case or type of inference task. If inference needs to be performed on thousands of observations, more time should be allocated to simulation and NPE should be chosen. If inference on only a few observations (or many different prior/simulator configurations) is required and/or the simulator is expensive, then NPE-PF is a great alternative. The stellar streams simulator is a good example of this. Attaining 100 simulations is feasible on a consumer-grade CPU (10 cores) in 5 hours, and NPE-PF achieves a comparable performance to NPE with 10,000 simulations, which would require 20 days (Table V1). In other words, NPE and NPE-PF serve different purposes.
>
> Third, we agree that NPE performance can be improved with a “well-selected embedding network”. However, training such a network requires a large amount of simulations. We discuss this situation in Appendix D.3. Specifically, in Fig D-4c, we compare NPE-PF with NPE using a tailored 1D CNN embedding net (Lotka Voltera [long] task with a 300 dim. time series), where NPE-PF still outperforms NPE.
>
> In summary, we believe that our experiments, both new and previous, conducted in various settings, constitute a "principled analysis of NPE-PF's diminishing advantages over standard NPE methods". Furthermore, we have already addressed several of the aforementioned “practical considerations” in the Appendix. However, we acknowledge that we did not communicate some of these aspects effectively throughout our manuscript and will address this in an updated version.
>
> **A3. Analysis of computational cost.**
>
> We agree with the reviewer that providing a comprehensive evaluation of the overall runtime is important. For this reason, we have included an overview in the main manuscript (Fig. 2b). However, we also agree that this is insufficient for assessing the "boundary beyond which one might consider a computationally faster method". To address this, we have now added a comprehensive runtime ablation section on data dimensionality and simulation size for NPE-PF.
>
> For results on runtime across different scenarios we refer to the response to Reviewer wFAY Table w2. To summarize the runtime analysis for NPE-PF:
> - The compute time grows with an increasing number of simulations and parameter dimensions, as expected.
> - In lower dimensional cases the compute time to acquire a 10k posterior samples ranges between 4-23 seconds across all context sizes.
> - In higher dimensional cases the compute time can grow to 1-2 minutes for context-sizes of 100/1000 and up to 8 minutes for 10000.
>
> While these inference times are substantially slower than standard NPE, the choice between both methods requires careful consideration of various factors, as discussed in **A1**.
>
>
> **Q1. Inheriting TabPFN properties.**
>
> We appreciate the reviewer’s suggestion to "characterise when, or if, NPE-PF struggles systematically". Indeed, one of our original objectives was to identify configurations that might present a challenge to NPE-PF. To this end, Section D.5 examines performance under extreme, heavy-tailed noise models. Across all noise scenarios that were tested, NPE-PF proved to be remarkably robust.
>
> Although we did not identify any systematic failure modes, we found that NPE-PF performs particularly well on simulators with an underlying graphical structure and conditional dependencies. We attribute this to the fact that TabPFN was pre-trained using random structural causal models (SCMs). See the response to Reviewer wFAY (**Q3**) for a more detailed discussion.
>
>
> **Q2. Parameter permutations in 31D example.**
>
> Testing the reliability of NPE-PF in high-dimensional spaces is inherently challenging. For the 31dim pyloric network, the total number of possible permutations of parameter dimensions is greater than $10^{33}$. Moreover, evaluating each candidate permutation requires expensive simulation, making exhaustive exploration impractical. To gain insight despite these constraints, we performed an ablation study on 50 distinct random permutations using the context set identified by the final‑round posterior. We compute the presented metrics with 1000 simulations. We found that the energy score varies between 0.081 - 0.094 and the valid rates between 96% - 98% (each the 5% and 95% quantile). This is in line with what we found using the default permutation in the paper (Appendix D.6, Fig. D-7). We note that these results must be interpreted with caution. There may be permutations for which NPE-PF performs significantly better or worse. However, identifying these permutations is very difficult due to the large number of possible permutations.
>
> [1] Hermans et al. "A trust crisis in simulation-based inference? Your posterior approximations can be unfaithful", 2021
> [2] Gloeckler et al. "All-in-one simulation-based inference", 2024

---

> > ### Comment · Reviewer_Vbbm · 2025-08-07
> >
> > I thank the authors for their thorough rebuttal and providing additional
> > experimental results. I have additionally read the other reviews/rebuttals and
> > am encouraged by the supplemental evaluations and clarifications. Most
> > of my concerns have been sufficiently addressed, from the additional tasks
> > reported in A.1 to the extended discussions in A.2/A.3. As a result, I maintain
> > my positive inclination toward acceptance, and am happy with the paper's
> > experimental validation, presentation, and discussion of limitations.

---

> > > ### Author Response · Authors · 2025-08-07
> > >
> > > Thank you for taking the time to engage with our work so thoroughly and for your positive response. Your feedback was valuable in helping us strengthen our work e.g. by extending our experimental validation.

---

### Note · Authors · 2025-08-11

We sincerely thank all reviewers for their thoughtful feedback. Based on this feedback, we here provide a summary of new results and revisions:

### **New experimental results**
Thanks to the reviewers constructive suggestions, we have performed several new experiments:
- **Analysis of computational cost (Vbbm, wFAY, BTVU)**: We analyzed the computational cost of NPE-PF across all relevant dimensions (number of simulations, dimensionality of observation and parameter space).
- **Additional tasks (Vbbm, BTVU)**: Beyond the standard benchmarks, we included cross-disciplinary and higher dimensional tasks.
- **Hyperparameter sweep for NPE baseline (BTVU)**: An NPE hyperparameter sweep showed beating the benchmark defaults is not easy; NPE-PF still performed better than NPE + Sweep (despite the large computational cost of sweeping).
- **Ablation study on the filter size (wFAY)**:  Our experiments showed that the default filter size of $10^4$ always works well, but can potentially be decreased to reduce computational cost (with small drops in performance).
- **Permutation analysis on the 31D parameter space of the pyloric experiment (Vbbm, RF8X)**: We analyzed the sensitivity to the autoregressive order in this high-dimensional task and observed only minor influence, which aligns with our benchmark results.


### **Additions and improvements to the text**

Based on reviewers' thoughtful feedback, we will include:

- A careful discussion of (TS)NPE-PF's computational cost in comparison to baseline methods, which will allow users to make an informed choice depending on their simulation budget and application (Vbbm, wFAY, BTVU).
- Improved discussion and interpretation of our results to better communicate the insights into when and where NPE-PF tends to work well (i.e., on "structured" simulators) (Vbbm, wFAY).
- Full results for the additional tasks in the Appendix  (Vbbm, BTVU).
- A better discussion on the role of parameter ordering and filter size (including our new results) (Vbbm, wFAY, RF8X).

These new results and revisions will substantially strengthen our manuscript, and we again thank the reviewers for their thoughtful feedback, constructive suggestions, and positive evaluation of our work.

---

### Decision · Program_Chairs · 2025-09-17

**Decision:**

Accept (poster)

**Comment:**

The paper develops a method for simulation-based inference that leverages a tabular foundation model to reduce the simulation burden. The key idea is to use TabFPN in an autoregressive factorization of the posterior. The resulting method can significantly reduce the number of simulations needed to obtain similarly good posterior approximations compared to other SBI methods.

Reviewers recognized the potential of using foundation models for SBI and found the idea to be simple, clever, and potentially impactful. They appreciated the details required to make it work. They thought the evidence was convincing within the scope of problems evaluated.

A weakness raised by several reviewers the lack of a clear discussion about total runtime complexity compared to alternatives, with one reviewer commenting that the tone of the paper might be a bit too optimistic by skirting this detail. The authors provided more information in the rebuttal. It is recommended to include this information and make this discussion more prominent in the revision.

Another weakness mentioned by multiple reviewers was the relative simplicity of problems in the benchmark. The evidence could be strengthened by using more complex real tasks. The authors provided more results in the rebuttal that strengthened the evidence.

Overall, the paper was well received and proposes an idea that may advance SBI and be used by other researchers.